# Breaking Class Barriers:
# Efficient Dataset Distillation
# via Inter-Class Feature Compensator

**Xin Zhang**[1,2]    **Jiawei Du**[1,2]   **Ping Liu**[3]   **Joey Tianyi Zhou**[1,2]✉

[1]Centre for Frontier AI Research, Agency for Science, Technology and Research, Singapore
[2]Institute of High Performance Computing, Agency for Science, Technology and Research, Singapore
[3]University of Nevada, Reno
{zhangx7, dujw, Joey_Zhou}@cfar.astar.edu.sg   pingl@unr.edu

## Abstract

Dataset distillation has emerged as a technique aiming to condense informative features from large, natural datasets into a compact and synthetic form. While recent advancements have refined this technique, its performance is bottlenecked by the prevailing class-specific synthesis paradigm. Under this paradigm, synthetic data is optimized exclusively for a pre-assigned one-hot label, creating an implicit class barrier in feature condensation. This leads to inefficient utilization of the distillation budget and oversight of inter-class feature distributions, which ultimately limits the effectiveness and efficiency, as demonstrated in our analysis. To overcome these constraints, this paper presents the Inter-class Feature Compensator (INFER), an innovative distillation approach that transcends the class-specific data-label framework widely utilized in current dataset distillation methods. Specifically, INFER leverages a Universal Feature Compensator (UFC) to enhance feature integration across classes, enabling the generation of multiple additional synthetic instances from a single UFC input. This significantly improves the efficiency of the distillation budget. Moreover, INFER enriches inter-class interactions during the distillation, thereby enhancing the effectiveness and generalizability of the distilled data. By allowing for the linear interpolation of labels similar to those in the original dataset, INFER meticulously optimizes the synthetic data and dramatically reduces the size of soft labels in the synthetic dataset to almost zero, establishing a new benchmark for efficiency and effectiveness in dataset distillation. In practice, INFER demonstrates state-of-the-art performance across benchmark datasets. For instance, in the `ipc = 50` setting on ImageNet-1k with the same compression level, it outperforms SRe2L by 34.5% using ResNet18. Codes are available at https://github.com/zhangxin-xd/UFC.

# 1 Introduction

The remarkable success of Deep Neural Networks (DNNs) (Yang et al., 2024; Cai et al., 2024; Zheng et al., 2024; Dosovitskiy et al., 2021) in recent years can largely be attributed to their ability to extract complex and representative features from vast real-world data (Gao et al., 2020; Benenson et al., 2019). However, the extensive data requirements for training DNNs pose significant challenges. These challenges include not only the time-consuming training process (Liu et al., 2021; Touvron et al., 2021; Tolstikhin et al., 2021), but also the substantial costs associated with data storage and computational resources (Kaplan et al., 2020; Hoffmann et al., 2022).

In response to the rapid growth of computational and storage demands in training DNNs, dataset distillation (DD) (Sachdeva & McAuley, 2023; Lei & Tao, 2023; Yu et al., 2023; Liu & Du, 2025) has emerged as an effective solution. Dataset distillation condenses essential features from extensive datasets into a compact, synthetic form, allowing models to maintain comparable performance levels with fewer resources (Wang et al., 2018). Recent advancements in dataset

---

✉ represents the corresponding author.

distillation encompass techniques such as gradient matching (Zhao et al., 2021; Zhao & Bilen, 2021; Lee et al., 2022b; Shin et al., 2023), trajectory matching (Cazenavette et al., 2022; Cui et al., 2023; Du et al., 2023a;b), data factorization (Liu et al., 2022; Kim et al., 2022; Wei et al., 2023; Shin et al., 2024), and kernel ridge regression (Nguyen et al., 2020; 2021; Loo et al., 2022). These approaches have significantly enhanced dataset distillation by compressing dense knowledge into single data instances. Despite the diversity of these methods, most of them adhere to a uniform paradigm: each synthetic data instance is class-specific and optimized exclusively for a pre-assigned one-hot label.

While this "one label per instance" paradigm aligns with the traditional data-label pair structure of original datasets, it presupposes that the most effective encapsulation of a dataset's knowledge can be achieved through individual

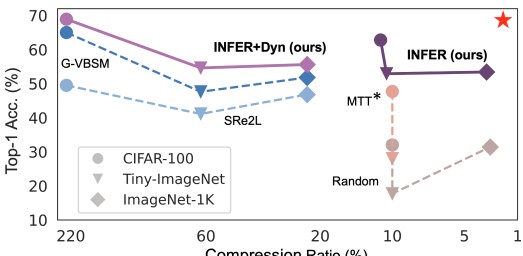

Figure 1: Performance vs. compression ratio of SOTA dataset distillation methods (G-VBSM (Shao et al., 2024), SRe2L (Yin et al., 2024), MTT (Cazenavette et al., 2022)) on three benchmarks. Performance is measured as the Top-1 accuracy of ResNet-18 (ConvNet128 for MTT) on the respective validation sets, trained from scratch using synthetic datasets. The compression ratio, including the additional soft labels, is the proportion of the distilled dataset size to the original dataset size. The star indicates optimal performance.

instances representing discrete class identities. When the amount of synthetic data is limited, this class-specific paradigm benefits distillation by encouraging synthetic data to condense the most distinctive features of each class. However, as more synthetic instances are assigned to the same class, they tend to capture significant but similar features rather than diversifying to include unique, rarer features. This phenomenon, known as "feature duplication" (Jiang et al., 2022; Kim et al., 2022; Cazenavette et al., 2022), leads to *Inefficient Utilization of the Distillation Budget*, thereby limiting the creation of a more diverse and comprehensive synthetic representation.

Another critical downside of the class-specific synthesis paradigm is the *"Oversight of Inter-Class Features"*. By focusing on distinctive class-specific characteristics under the "one label per instance" approach, implicit **"Class Barriers"** are created between classes. These class barriers prevent the synthetic data instances from capturing inter-class features that bridge different classes in the original dataset. This oversight, inhibits the formation of thin and clear decision boundaries among classes, which are essential for models to generalize well across complex scenarios. We demonstrate the visualization of frormed decision boundaries in Figure 2.

Recognizing the aforementioned limitations, we introduce a novel paradigm for dataset distillation, termed the Inter-class Feature compEnsatoR (INFER). Unlike traditional methods that follow the "one instance for one class" paradigm and generate separate synthetic instances for each class, INFER pioneers a "one instance for ALL classes" paradigm by introducing a Universal Feature Compensator (UFC). The UFC, designed to reflect the general representativeness across all classes of the original dataset, depreciates the importance of pre-assigned labels. This feature enables UFC to compensate for inter-class features while INFER randomly incorporates a few natural data instances to enhance intra-class features. Notably, INFER integrates one UFC with multiple natural data instances from different classes through a simple additive process without auxiliary generator networks (Liu et al., 2022), allowing the generation of multiple synthetic instances from a single input. This "one instance for ALL classes" paradigm significantly enhances the efficiency of the distillation budget.

Furthermore, we have meticulously designed the optimization of UFCs to encompass inter-class features. This optimization makes synthetic instances generated from UFCs compatible with MixUp data augmentation, which promotes inter-class interactions and aids in forming thin, clear decision boundaries among classes. Prior works applying MixUp to synthetic data (Yin et al., 2024) involve dynamic generating and storing an extensive amount of soft labels, which can increase storage requirements up to 30-fold. INFER, however, dramatically eliminates the need for such extensive soft label storage by elegantly adopting the linear interpolation of labels used with natural datasets, decreasing the storage requirement by 99.3%. This enhancement not only preserves the distillation budget but also streamlines the entire training process, underscoring INFER's efficiency and effectiveness. Notably, INFER achieves 53.3% accuracy on ImageNet-1k with a ResNet-50 model, training solely on the synthetic dataset, which is only 4.04% the size of the original ImageNet-1k.

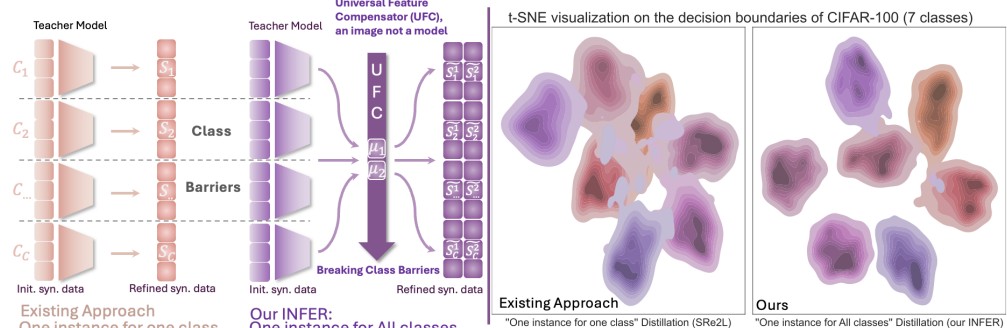

Figure 2: **Left:** Overview of dataset distillation paradigms. The first illustrates the traditional "one instance for one class" approach, where each instance is optimized exclusively for its pre-assigned label, creating implicit class barriers. The second illustrates our INFER method, designed for "one instance for ALL classes" distillation. **Right:** t-SNE visualization of the decision boundaries between the traditional approaches (*i.e.*, SRe2L (Yin et al., 2024)) and our INFER approach. We randomly select seven classes from CIFAR-100 dataset for the visualization. INFER forms thin and clear decision boundaries among classes, in contrast to the chaotic decision boundaries of the traditional approach.

This performance, which does not require dynamically generated soft labels, outperforms existing approaches in both accuracy and compression ratio.

Our contribution can be summarized as follows:

- We rethink the prevailing "one label per instance" paradigm that exclusively optimizes each synthetic data instance for a specific class. Through empirically analysis, we identify and address its two main limitations: inefficient utilization of the distillation budget and oversight of inter-class features.
- To overcome these issues, we introduce a new paradigm INFER, for "one instance for all classes" dataset distillation. Our INFER incorporates a novel Universal Feature Compensator (UFC) to efficiently condense and integrate features across multiple classes. Extensive experiments across CIFAR, tiny-ImageNet and ImageNet-1k datasets demonstrate the state-of-the-art performance of INFER.

## 2   PRELIMINARIES AND RELATED WORKS

The precursor to dataset distillation in condensing of datasets is coreset selection (Bachem et al., 2017; Chen et al., 2010; Har-Peled & Kushal, 2005; Sener & Savarese, 2018; Xin et al., 2024). This method involves selecting a coreset of the original dataset that ideally contains the entire representativeness of the population. However, this approach encounters a significant performance drop when compression ratio is small. A plausible explanation for this could be the low density of representativeness within natural data instances. Therefore, dataset distillation seeks to synthesize data instances with densely packed features. We begin with a brief formulation of dataset distillation.

**Problem Formulation.** Assume we are given a natural and large dataset $\mathcal{T} = \{(\boldsymbol{x}_i, \boldsymbol{y}_i)\}_{i=1}^{|\mathcal{T}|}$, where each element $\boldsymbol{x}_i \in \mathbb{R}^d$ is drawn i.i.d. from a natural distribution $\mathcal{D}$, and the class label $\boldsymbol{y}_i \in \mathcal{Y} = \{0, 1, \ldots, C-1\}$ with $C$ representing the number of classes. Dataset distillation aims to synthesize a small dataset $\mathcal{S} = \{(\boldsymbol{s}_i, \boldsymbol{y}_i)\}_{i=1}^{|\mathcal{S}|}$, where $\boldsymbol{s}_i \in \mathbb{R}^d$ and $\boldsymbol{y}_i \in \mathcal{Y}$, to serve as an approximate solution to the following optimization problem:

$$\mathcal{S} = \underset{\mathcal{S} \subset \mathbb{R}^d \times \mathcal{Y}}{\arg\min} \; \underset{(\boldsymbol{x}, \boldsymbol{y}) \sim \mathcal{D}}{\mathbb{E}} \left[ \ell\left(f_{\theta_{\mathcal{S}}}, \boldsymbol{x}, \boldsymbol{y}\right) \right], \tag{1}$$

where $\theta_{\mathcal{S}}$ represents the converged weights trained with $\mathcal{S}$, and $\ell$ is the loss function. The class label $\boldsymbol{y}_i$ is typically a pre-assigned one-hot label (Zhao & Bilen, 2023; Zhao et al., 2021; Zhao & Bilen, 2021; Jiang et al., 2022; Kim et al., 2022; Cazenavette et al., 2022) to encourage $\mathbf{s}_i$ to exhibit more distinct, class-specific features.

---

Compression Ratio (CR) = synthetic dataset size / original dataset size (Cui et al., 2022). A lower ratio indicates a more condensed dataset.

Pioneering our approach, Wang et al. (Wang et al., 2018) is the first to propose DD, a method that optimizes $\mathcal{S}$ directly after substituting $\mathcal{D}$ with $\mathcal{T}$ in Equation 1. Due to the limited guidance, this approach often leads to suboptimal performance, prompting the development of gradient-matching methods (Zhao et al., 2021; Zhao & Bilen, 2021; Lee et al., 2022b; Shin et al., 2023). These methods improve supervision by aligning the models' gradients. The success of gradient-matching has inspired further research into matching the trajectory of gradients (Cazenavette et al., 2022; Cui et al., 2023; Du et al., 2023a;b), yielding even better performance. Despite these advancements, most of current dataset distillation methods still primarily focus on generating class-specific synthetic instances. This ongoing adherence to the class-specific paradigm not only constrains the efficiency of the distillation budget but also results in the neglect of critical inter-class features. These limitations drive our investigation into a new paradigm, aimed at developing a more efficient and effective dataset distillation solution.

**Distillation Budget Consistency.** As our INFER depreciates the class-specific paradigm, it becomes necessary to establish clear criteria for maintaining consistency in the distillation budget. Traditional methods adopt Images Per Class (IPC) as described by (Wang et al., 2018; Zhao et al., 2021; Cazenavette et al., 2022), such that $|\mathcal{S}| = \mathtt{ipc} \times C$. This approach provides a uniform criterion across various datasets. Therefore, we continue to employ IPC as the criterion to measure the distillation budget.

However, IPC does not account for the auxiliary generator (Liu et al., 2022; Lee et al., 2022a) or additional soft labels (Yin et al., 2024) that are used to enhance the performance of a synthetic dataset, resulting in an asymmetric advantage compared to methods that solely utilize the data-label pair. Therefore, we also employ the compression ratio, as described by (Cui et al., 2022), to measure the distillation budget. The total bit count of any auxiliary modules and soft labels will be considered part of the synthetic dataset. To compute the compression ratio, we divide the total bit count of the synthetic dataset, by that of the original dataset, *i.e.*, $\mathtt{CR} = $ synthetic dataset size/original dataset size.

# 3 METHODOLOGY

In this section, we introduce our novel distillation paradigm, INFER. We begin by detailing the limitations associated with the class-specific distillation paradigm, highlighting inefficiencies and oversight of inter-class feature distributions. Following this, we describe the Universal Feature Compensator (UFC), the cornerstone of our methodology, designed to integrate inter-class features. Lastly, we discuss our approach to augmenting synthetic datasets, which aims to facilitate the formation of thin and clear decision boundaries among classes.

## 3.1 LIMITATIONS IN CLASS-SPECIFIC DISTILLATION

Wang et al. (Wang et al., 2018) established the general approach to solve $\mathcal{S}$ as outlined in Equation 1: Each synthetic data instance $s_i$ is assigned a one-hot label $y_i$ and optimized to capture intra-class features by minimizing $\sum_{(x,y)\in\mathcal{T}} \ell\left(f_{\theta_{\mathcal{S}}}, x, y\right)$. Unlike Equation 1, $\mathcal{D}$ is replaced by $\mathcal{T}$, given that $\mathcal{D}$ is inaccessible and $\mathcal{T} \sim \mathcal{D}$. Initially, this class-specific design achieved progress in the early stages. However, as dataset distillation research has evolved, two major limitations become prominent, compelling a rethinking of this design.

**Inefficient Utilization of Distillation Budget.** Recent advancements in dataset distillation (Yin et al., 2024; Loo et al., 2022; Cui et al., 2023) have enabled individual synthetic data instances to capture more features specific to a class, particularly notable in highly compressed scenarios where $\mathtt{ipc} = 1$ (Cui et al., 2023; Loo et al., 2022). However, as $\mathtt{ipc}$ increases, additional synthetic data instances tend to capture distinctive yet duplicated intra-class features, leading to redundancy within the synthetic dataset. This redundancy explains the marginal performance gains observed in less compressed scenarios ($\mathtt{ipc} = 50$). SeqMatch (Du et al., 2023b) and DATM (Guo et al., 2023) addressed this redundancy by dividing the synthetic data into several subsets optimized diversely. We validate this hypothesis through experiments shown in Figure 4 (a). Ideally, newly optimized synthetic data instances should capture rare and diversifying features that complement the distinctive class-specific features.

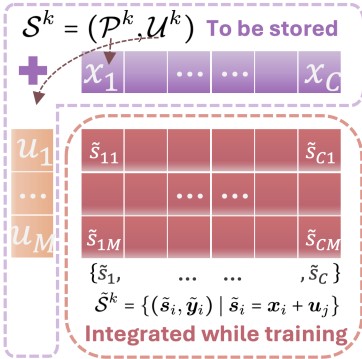

Figure 3: Illustration of the integration process between Universal Feature Compensators (UFCs) and natural data instances as described in Equation 2. The integration is performed through a simple addition process. Consequently, only the sets $\mathcal{S} = (\mathcal{P}^k, \mathcal{U}^k)$ need to be stored as the synthetic dataset. The synthetic dataset $\tilde{\mathcal{S}}^k$ is generated on-the-fly during training.

**Oversight of Inter-class Features.** The prevalent focus on optimizing synthetic instances for the most distinctive intra-class features often leads to the oversight of inter-class features. This oversight significantly limits the synthetic data's ability to represent the feature distributions that span across different classes, which is crucial for complex classification tasks. Consequently, the potential for these synthetic datasets to support the training of models that can generalize across varied scenarios may be significantly hampered. As depicted in Figure 2, this issue is evidenced by the chaotic decision boundaries formed by the "one label per instance" synthetic dataset, which neglects the inter-class features necessary for forming thin and clear decision boundaries.

These limitations drive our investigation into a new paradigm, aimed at developing a more efficient and effective dataset distillation solution.

### 3.2 UNIVERSAL FEATURE COMPENSATOR: BREAKING CLASS BARRIERS

The Universal Feature Compensators (UFCs), denoted as $\mathcal{U} = \{\boldsymbol{u}_i\}_{i=1}^{|\mathcal{U}|}$, forms the core of our novel INFER paradigm, designed specifically to address the inefficiencies and oversight inherent in the class-specific distillation approach.

**Design and Functionality.** The primary objective of designing the UFC is to enable the generation of multiple synthetic instances from a single compensator. To achieve this, our INFER divides the base synthetic dataset $\mathcal{S}$ into $K$ subsets, such that $\mathcal{S} = \mathcal{S}^1 \cup \mathcal{S}^2 \cup \cdots \cup \mathcal{S}^K$. Each base subset $\mathcal{S}^k$ consists of a pair $(\mathcal{P}^k, \mathcal{U}^k)$, where $\mathcal{U}^k$ represents the set of UFCs, and $\mathcal{P}^k \subset \mathcal{T}$ contains natural instances to be integrated with UFCs and $|\mathcal{P}^k| = C$. For actual model training on the synthetic dataset, we first integrates UFCs with natural data instances as follows:

$$\tilde{\mathcal{S}}^k = \{(\tilde{\boldsymbol{s}}_i, \tilde{\boldsymbol{y}}_i) \mid \tilde{\boldsymbol{s}}_i = \boldsymbol{x}_i + \boldsymbol{u}_j, \\ \text{for each } \boldsymbol{x}_i \in \mathcal{P}^k \text{ and each } \boldsymbol{u}_j \in \mathcal{U}^k\}, \tag{2}$$

where $\tilde{\mathcal{S}}^k$ represents the UFCs integrated synthetic dataset. This process is illustrated in Figure 3. In practice, multiple architectures participate in UFCs' generation. Assuming the number of architectures is $M$, *i.e.*, $|\mathcal{U}^k| = M$, each $\mathcal{S}^k$ can be multiplied approximately $M$ times by INFER. We repeat the intergration process described above $K$ times, once for each subset of $\mathcal{S}$. This structured approach allows INFER to utilize the distillation budget approximately **$M$ times** more efficiently compared to the class-specific paradigm.

**Optimization.** Under the INFER paradigm, each $\mathcal{P}^k$ contains exactly one instance for each class, randomly selected from the original dataset $\mathcal{T}$. Consequently, $\mathcal{P}^k$ effectively captures the intra-class features with minimal duplication. In contrast, the UFCs are designed to optimize the capture of inter-class features. Therefore, the elements in $\mathcal{P}^k$ uniformly cover each class, allowing the element $\mathcal{U}^k$ to focus on capturing inter-class features by solving the follow minimization problem:

$$\underset{\boldsymbol{u}_j \in \mathbb{R}^d}{\arg\min} \sum_{(\boldsymbol{x}_i, \boldsymbol{y}_i) \in \mathcal{P}^k} \left[ \ell\left(f_{\theta_\mathcal{T}}, \boldsymbol{x}_i + \boldsymbol{u}_j, \boldsymbol{y}_i\right) + \alpha \mathcal{L}_{\mathrm{BN}}\left(f_{\theta_\mathcal{T}}, \boldsymbol{x}_i + \boldsymbol{u}_j\right) \right], \tag{3}$$

$$\text{where} \quad \mathcal{L}_{\mathrm{BN}}\left(f_{\theta_\mathcal{T}}, \boldsymbol{x}_i + \boldsymbol{u}_j\right) = \sum_l \|\mu_l(\tilde{\mathcal{S}}_j^k) - \mu_l(\mathcal{T})\|_2 \\ + \sum_l \|\sigma_l^2(\tilde{\mathcal{S}}_j^k) - \sigma_l^2(\mathcal{T})\|_2. \tag{4}$$

we define $\tilde{\mathcal{S}}_j^k = \{(\tilde{\boldsymbol{s}}_i, \tilde{\boldsymbol{y}}_i) \mid \tilde{\boldsymbol{s}}_i = \boldsymbol{x}_i + \boldsymbol{u}_j, \text{ for each } \boldsymbol{x}_i \in \mathcal{P}^k\}$ which is the generated synthetic dataset by integrating $\boldsymbol{u}_j$ with the corresponding $\mathcal{P}^k$. $\mathcal{L}_{\mathrm{BN}}$ is the BN loss inspired by SRe2L (Yin et al., 2024) to regularize the values of generated $\tilde{\boldsymbol{s}}_i$ to fall within the same normalization distribution

as those in $\mathcal{T}$. Intuitively, $\boldsymbol{u}_j$ serves as the feature compensator for the natural instances in $\mathcal{S}$, carrying universal inter-class features that are beneficial for classification.

---

**Algorithm 1** Distillation on synthetic dataset via Inter-class Feature Compensator (INFER)

---

**Require:** Target dataset $\mathcal{T}$; Number of subsets $K$; Number of classes $C$; $M$ networks with different architectures:$\{f^1, f^2, \cdots, f^M\}$.
1: Initialize $\mathcal{S} = \{\}$
2: **for** $k = 1$ to $K$ **do**
3:  Initialize subset $\mathcal{P}^k = \{\}$, the UFCs set $\mathcal{U}^k = \{\}$, and the static labels set $\mathcal{Y}^k = \{\}$
4:  Randomly select $C$ instances, one for each class, to form $\mathcal{P}^k$, such that:
5:  $\mathcal{P}^k = \{(\boldsymbol{x}_i, \boldsymbol{y}_i) \mid (\boldsymbol{x}_i, \boldsymbol{y}_i) \in \mathcal{T}$ and each $\boldsymbol{y}_i$ is unique in $\mathcal{P}^k\}$
6:  Initialize $\mathcal{U}^k$ with zeros, where each $\boldsymbol{u}_i$ has the same dimensions as $\boldsymbol{x}$:
7:  $\mathcal{U}^k = \{\boldsymbol{u}_j \mid \boldsymbol{u}_j = \boldsymbol{0}_{\dim(\boldsymbol{x})}$, for $j = 1, \ldots, M\}$
8:  **for** each $\boldsymbol{u}_j$ in $\mathcal{U}^k$ **do**
9:    ▷ Construct integrated synthetic instance $\tilde{\boldsymbol{s}}_i$
10:   Let $\tilde{\mathcal{S}}_j^k = \{(\tilde{\boldsymbol{s}}_i, \tilde{\boldsymbol{y}}_i) \mid \tilde{\boldsymbol{s}}_i = \boldsymbol{x}_i + \boldsymbol{u}_j$, for each $\boldsymbol{x}_i \in \mathcal{P}^k\}$
11:   **repeat**
12:     Optimize $\boldsymbol{u}_j$ to minimize loss defined in Equation 3
13:   **until** Converge
14:   Generate static soft labels $\tilde{\boldsymbol{y}}_i$ by Equation 6, $\mathcal{Y}^k = \mathcal{Y}^k \cup \{\tilde{\boldsymbol{y}}_i\}$
15:  **end for**
16:  $\mathcal{S}^k = \{\mathcal{P}^k, \mathcal{U}^k, \mathcal{Y}^k\}$
17:  $\mathcal{S} = \mathcal{S} \cup \{\mathcal{S}^k\}$
18: **end for**
**Ensure:** Synthetic dataset $\mathcal{S}$

---

### 3.3 ENHANCING SYNTHETIC DATA WITH INTER-CLASS AUGMENTATION

Although UFCs are encouraged to encapsulate more inter-class features, their limited size, especially compared to the size of the original dataset $\mathcal{T}$, restricts the breadth of features crucial for forming the decision boundaries of neural networks. Therefore, we also leverage MixUP (Zhang et al., 2018) as a technique to enhance inter-class augmentation.

Data augmentation (Shorten & Khoshgoftaar, 2019) has been well-developed and proven effective in training neural networks with natural datasets. However, the advancements in data augmentation do not generalize well to synthetic datasets generated through dataset distillation. DSA (Zhao & Bilen, 2021) designed an adapted augmentation method specialized for dataset distillation, but it is not as effective as standard data augmentation methods. SRe2L (Yin et al., 2024) applies MixUp (Zhang et al., 2018) to the synthetic dataset, achieving superior performance across many datasets. Unfortunately, the cost of applying MixUp in SRe2L is expensive, due to the massive volume of soft labels. The soft labels are dynamically generated for each augmented instance in each validation epoch, resulting enormous storage requirements or extra training efforts. For example, the synthetic dataset generated by SRe2L for ImageNet-1k requires 0.7GB to store synthetic images, but requires additional 25.9GB to store the soft labels.

Motivated by this, we aim to apply MixUp to our INFER model in the same way it is used in natural datasets, which we refer to as "static" soft labels. The linear interpolation of labels can be represented mathematically as:

$$
\begin{aligned}
&f_{\theta_{\mathcal{T}}}[\lambda \tilde{\boldsymbol{s}}_i + (1-\lambda)\tilde{\boldsymbol{s}}_j] \\
&\approx \lambda f_{\theta_{\mathcal{T}}}(\tilde{\boldsymbol{s}}_i) + (1-\lambda)f_{\theta_{\mathcal{T}}}(\tilde{\boldsymbol{s}}_j), \forall \tilde{\boldsymbol{s}}_i, \tilde{\boldsymbol{s}}_j \in \tilde{\mathcal{S}}
\end{aligned}
\tag{5}
$$

where $f_{\theta_{\mathcal{T}}}(\cdot)$ represents the logits output, $\lambda \sim \text{Beta}(\beta, \beta)$, and $\beta > 0$. To achieve this, we propose three improvements in INFER: (a) We use $\mathcal{P}^k$, a subset of natural datasets, for integration with UFCs because natural instances inherently follow the linear interpolation of labels. (b) We make $\mathcal{P}^k$ to span across all the classes, rather than limiting it to instances within the same class. As such, the optimized UFCs, $\mathcal{U}$, which are integrated with $\mathcal{P}^k$ for optimization, also embody the characteristic of linear label interpolation. (c) We employ $M$ neural networks with various architectures ($M = |\mathcal{U}^k|$)

Table 1: Comparison with SOTAs on CIFAR-10/100 and Tiny-ImageNet. Except for SRe2L (Yin et al., 2024), G-VBSM (Shao et al., 2024), and our INFER, all other methods use ConvNet128 for distillation. The distilled synthetic datasets are then evaluated on ConvNet128 and ResNet18. "IN-FER+Dyn" denotes the application of INFER using dynamically generated soft labels, as described in SRe2L (Yin et al., 2024). The best performers in each setting are highlighted in **red**.

| | | CIFAR-10 | | CIFAR-100 | | | Tiny-ImageNet | |
| --- | --- | --- | --- | --- | --- | --- | --- | --- |
| | `ipc` | 10 | 50 | 10 | 50 | 100 | 10 | 50 |
| ConvNet128 | Random | 31.0 ±0.5 | 50.6 ±0.3 | 14.6 ±0.5 | 33.4 ±0.4 | 42.8 ±0.3 | 5.0 ±0.2 | 15.0 ±0.4 |
| | DC (Zhao et al., 2021) | 44.9 ±0.5 | 53.9 ±0.5 | 25.2 ±0.3 | - | - | - | - |
| | DSA (Zhao & Bilen, 2021) | 52.1 ±0.5 | 60.6 ±0.5 | 32.3 ±0.3 | 42.8 ±0.4 | - | - | - |
| | KIP (Nguyen et al., 2021) | 62.7 ±0.3 | 68.6 ±0.2 | 28.3 ±0.1 | - | - | - | - |
| | RFAD (Loo et al., 2022) | **66.3** ±0.5 | 71.1 ±0.4 | 33.0 ±0.3 | - | - | - | - |
| | MTT (Cazenavette et al., 2022) | 65.4 ±0.7 | 71.6 ±0.2 | 39.7 ±0.4 | 47.7 ±0.2 | 49.2 ±0.4 | 23.2 ±0.2 | 28.0 ±0.3 |
| | SeqMatch (Du et al., 2023b) | 66.2 ±0.6 | **74.4** ±0.5 | **41.9** ±0.5 | 51.2 ±0.3 | - | 23.8 ±0.3 | - |
| | G-VBSM (Shao et al., 2024) | 46.5 ±0.7 | 54.3 ±0.3 | 38.7 ±0.2 | 45.7 ±0.4 | - | - | - |
| | **INFER** | 34.0 ±0.4 | 57.0 ±0.2 | 41.0 ±0.4 | **53.8** ±0.2 | **57.0** ±0.02 | 22.8 ±0.3 | 32.3 ±0.3 |
| | **INFER**+Dyn | 30.1 ±0.8 | 52.4 ±0.7 | 37.2 ±0.3 | 50.7 ±0.3 | 53.4 ±0.2 | **24.9** ±0.3 | **33.9** ±0.6 |
| ResNet18 | Random | 29.6 ±0.9 | 36.7 ±1.7 | 15.8 ±0.2 | 32.0 ±0.0 | 47.5 ±0.0 | 12.1 ±0.3 | 17.7 ±0.0 |
| | SRe2L (Yin et al., 2024) | 27.2 ±0.5 | 47.5 ±0.6 | 31.6 ±0.5 | 49.5 ±0.3 | - | - | 41.1 ±0.4 |
| | G-VBSM (Shao et al., 2024) | **53.5** ±0.6 | 59.2 ±0.4 | **59.5** ±0.4 | 65.0 ±0.5 | - | - | 47.6 ±0.3 |
| | **INFER** | 32.0 ±0.5 | 60.4 ±1.6 | 45.2 ±0.04 | 62.8 ±0.4 | 66.3 ±0.1 | 32.0 ±0.1 | 52.9 ±0.1 |
| | **INFER**+Dyn | 30.7 ±0.3 | **60.7** ±0.9 | 53.4 ±0.6 | **68.9** ±0.1 | **73.3** ±0.2 | **41.0** ±0.4 | **54.6** ±0.4 |

to relabel the generated synthetic data instance $\tilde{s}_i$ through averaging, *i.e.*,

$$\tilde{\boldsymbol{y}}_i = \frac{1}{M} \sum_m f_{\theta_\mathcal{T}}^m(\tilde{\boldsymbol{s}}_i). \tag{6}$$

By doing so, INFER does not require the dynamic soft labels for any combinations of $[\lambda \tilde{s}_i + (1 - \lambda)\tilde{s}_j]$, but only the static soft labels of $\tilde{s}_i, \tilde{s}_j$, which reduces the size of soft labels by up to 99.9%. Additionally, training on synthetic datasets can follow the same paradigm as training on natural datasets. More details of synthesizing and training $\mathcal{S}$ can be found in Algorithm 1 and Algorithm 2.

## 4 EXPERIMENTS

To evaluate the effectiveness of our proposed INFER distillation paradigm, we conducted a series of experiments across multiple benchmark datasets and compared our results with several state-of-the-art approaches. In this section, we provide details on the experimental setup, the datasets used, and the results obtained. We summarize our main results in Table 1 and Table 2. Following this, we perform ablation studies to assess the impact of individual components of our method. All experiments were conducted using two Nvidia 3090 GPUs and one Tesla A-100 GPU.

### 4.1 EXPERIMENTAL SETUP

**Baselines and Datasets.** We conduct the comparison with several representative distillation methods, including Random, DC (Zhao et al., 2021), DSA (Zhao & Bilen, 2021), KIP (Nguyen et al., 2021), RFAD (Loo et al., 2022), MTT (Cazenavette et al., 2022), SeqMatch (Du et al., 2023b), G-VBSM (Shao et al., 2024) and SRe2L (Yin et al., 2024). This evaluation is performed on four popular classification benchmarks, including CIFAR-10/100 (Krizhevsky et al., 2009), Tiny-ImageNet (Le & Yang, 2015), and ImageNet-1k (Deng et al., 2009).

**Implementation Details.** Our INFER uses $M = 4$, meaning it employs four different architectures for optimizing UFCs: ResNet18 (He et al., 2016), MobileNetv2 (Sandler et al., 2018), Efficient-

Table 2: Comparison with SOTAs on ImageNet-1k. SRe2L (Yin et al., 2024), G-VBSM (Shao et al., 2024), and our INFER use ResNet18 for distillation. The distilled synthetic datasets are then evaluated on ResNet18, 50, and 101. "INFER+Dyn" denotes the application of INFER using dynamically generated soft labels, as described in SRe2L (Yin et al., 2024). We also evaluate SRe2L and G-VBSM under the same compression ratio using our static labeling strategy, denoted as {}*. The best performers in each setting are highlighted in **red**.

| | Compression Ratio | | ResNet18 | | ResNet-50 | | ResNet-101 | |
| ipc | 10 | 50 | 10 | 50 | 10 | 50 | 10 | 50 |
|---|---|---|---|---|---|---|---|---|
| Random | 0.78% | 3.90% | 10.5 ±0.4 | 31.4 ±0.3 | 9.3 ±0.3 | 31.5 ±0.2 | 10.0 ±0.4 | 33.1 ±0.1 |
| SRe2L* (Yin et al., 2024) | 0.81% | 4.04% | 9.8 ±0.1 | 17.3 ±0.5 | 8.7 ±0.3 | 17.2 ±0.4 | 8.8 ±0.2 | 15.8 ±0.2 |
| G-VBSM* (Shao et al., 2024) | 0.81% | 4.04% | 11.9 ±0.2 | 32.9 ±0.1 | 14.5 ±0.2 | 38.1 ±0.2 | 13.9 ±0.1 | 38.9 ±0.4 |
| **INFER** | 0.81% | 4.04% | **28.7** ±0.2 | **51.8** ±0.2 | **26.9** ±0.3 | **53.3** ±0.3 | **26.5** ±0.1 | **52.2** ±0.3 |
| SRe2L (Yin et al., 2024) | 4.53% | 22.67% | 21.3 ±0.6 | 46.8 ±0.2 | 28.4 ±0.1 | 55.6 ±0.3 | 30.9 ±0.1 | 60.8 ±0.5 |
| G-VBSM (Shao et al., 2024) | 4.53% | 22.67% | 31.4 ±0.5 | 51.8 ±0.4 | 35.4 ±0.8 | 58.7 ±0.3 | 38.2 ±0.4 | **61.0** ±0.4 |
| **INFER**+Dyn | 4.53% | 22.67% | **36.3** ±0.3 | **55.6** ±0.2 | **38.3** ±0.5 | **63.4** ±0.3 | **38.9** ±0.5 | 60.7 ±0.1 |

NetB0 (Tan & Le, 2019), and ShuffleNetv2 (Ma et al., 2018). When distilling ImageNet-1k, only the first three architectures ($M = 3$) are involved. For reproducibility, the hyperparameter settings for the experimental datasets—CIFAR-10/100, Tiny-ImageNet, and ImageNet-1k, are provided in Appendix A.3. These settings generally follow SRe2L (Yin et al., 2024), with the sole modification being a proportional reduction in the validation epoch number for the dynamic version to ensure fair comparison. All other critical hyperparameters remain unchanged.

**Consistant Distillation Budget.** As we stated in Section 2, the baselines SRe2L (Yin et al., 2024) and G-VBSM (Shao et al., 2024) use the dynamic generated soft labels from a teacher model in each validation epoch for enhanced performance. However, these additional dynamic soft labels are not considered in the "Images Per Class (IPC)" distillation budget. Therefore, we also adopt the compression ratio (CR = synthetic dataset size/original dataset size (Cui et al., 2022)) for consistant distillation budget. As the size of the soft labels is proportional to the number of validation epochs, we only report the CR in ImageNet-1k dataset, as shown in Table 2.

Our INFER also employs the soft labels as shown in Equation 6. However, INFER only stores one soft label per instance, which equals **one** epoch dynamic soft labels. We term it as static soft labels in contrast to the dynamic soft labels generated across every validation epochs. Therefore, INFER reduces the size of soft labels by up to 99.3% in ImageNet-1k dataset (from 300 epoch to 1 epoch). We also implement the dynamic soft labels under INFER, denoted as "INFER+Dyn". As increasing the validation epoch can improve the performance, but it incurs a larger size of soft labels. Another point to note is, to ensure fair comparison, we reduce the validation epoch to $\frac{1}{M}$, as INFER generates $M$-fold synthetic instances by integrating UFC ($M$ is 3 for ImageNet-1k). Lastly, we also average the number of UFCs into ipc and adjust $K = \lfloor ipc \times \frac{C}{C+M} \rfloor$ for fair comparison.

## 4.2 MAIN RESULTS

Performance results on CIFAR-10/100 and Tiny-ImageNet are summarized in Table 1. While most previous methods rely on ConvNet128 due to resource constraints, SRe2L (Yin et al., 2024), G-VBSM (Shao et al., 2024), and our INFER method use ResNet18 for synthesis. INFER significantly outperforms competitors. With ipc=50, ResNet18 trained on the distilled CIFAR-100 achieves 68.9% and 62.8% accuracy, surpassing SRe2L by 19.4% and 13.3%, respectively. INFER also outperforms G-VBSM by 5.3% on Tiny-ImageNet. Despite reducing 99.3% of dynamic soft labels, INFER still outperforms methods using dynamic labels. Static labels suffice for simpler networks and datasets, but dynamic labels provide stronger supervision for complex architectures.

Table 2 provides a detailed performance comparison between SRe2L (Yin et al., 2024), the pioneering method for scaling dataset distillation to large datasets like ImageNet-1k, its extension G-VBSM (Shao et al., 2024), and our proposed INFER approach. These methods use ResNet18 for dataset distillation, and their performance is evaluated on three architectures: ResNet18, ResNet50, and ResNet101. The table highlights the advantages of our INFER method, which consistently outperforms SRe2L across all evaluation settings. Notably, when distilling datasets with ipc=50,

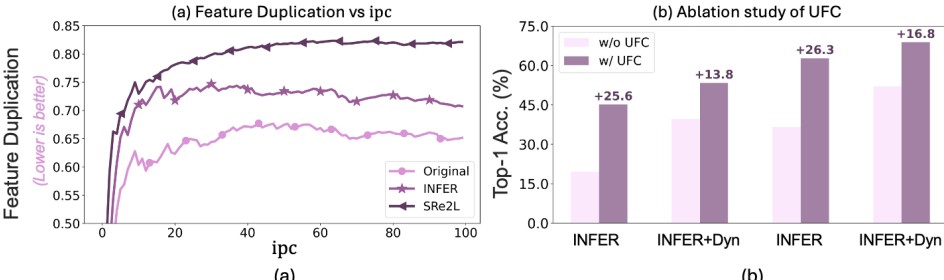

Figure 4: **Left:** The change in feature duplication with the increase of `ipc`. To measure the level of feature duplication, we employ the averaged cosine similarities between each pair of synthetic data instances within the same class. Therefore, a greater value represents higher feature duplication, as SRe2L (Yin et al., 2024) shows. In contrast, our INFER obtains a lower feature duplication, which is closer to the level observed in natural datasets. **Right:** The ablation study of UFC. The first two groups are under the `ipc = 10` setting, while the other two are under `ipc = 50`. The **purple** annotations indicate the performance gains contributed by our UFC.

Table 3: Ensemble of architectures for UFC generation. "R", "M", "E", and "S" represent ResNet18 (He et al., 2016), MobileNetv2 (Sandler et al., 2018), EfficientNetB0 (Tan & Le, 2019), and ShuffleNetV2 (Ma et al., 2018), respectively. ✔ indicates the network architectures participating in UFC generation. ↑ denotes the performance gain contributed by the current ensembles compared with the baseline (only ResNet18). These experiments are conducted on CIFAR-100 dataset.

| R | M | E | S | ipc = 10 | | | | ipc = 50 | | | |
|---|---|---|---|---|---|---|---|---|---|---|---|
| | | | | INFER | ↑ | INFER+Dyn | ↑ | INFER | ↑ | INFER+Dyn | ↑ |
| ✔ | | | | 40.9 ±0.1 | + 0.0 | 38.1 ±1.7 | + 0.0 | 59.7 ±0.2 | + 0.0 | 65.3 ±0.2 | + 0.0 |
| ✔ | ✔ | | | 43.2 ±0.2 | + 2.3 | 46.2 ±0.4 | + 8.1 | 61.2 ±0.04 | + 1.5 | 67.3 ±0.1 | + 2.0 |
| ✔ | ✔ | ✔ | | 44.8 ±0.5 | + 3.9 | 50.6 ±0.7 | + 12.5 | 61.7 ±0.1 | + 2.0 | 68.3 ±0.2 | + 3.0 |
| ✔ | ✔ | ✔ | ✔ | 45.2 ±0.04 | + 4.3 | 53.4 ±0.6 | + 16.3 | 62.8 ±0.4 | + 3.1 | 68.9 ±0.1 | + 3.6 |

INFER achieves a substantial 5.0% performance improvement on ResNet18 while maintaining an extremely compact synthetic dataset—only 4.04% of the original size. When matched to the same compression ratio, the performance gap further widens to 34.5%. This result showcases the superior efficiency and effectiveness of INFER in large-scale dataset distillation, particularly in compressing datasets while maintaining performance.

### 4.3 ABLATION STUDY

**Compensator Generation.** We examine the effectiveness of the proposed UFC. As shown in Figure 4 (b), regardless of the `ipc` setting and the labeling strategy, our UFC significantly enhances the quality of the distilled dataset. For example, the Top-1 classification accuracy of the model trained with static labels is improved by 25.6% with `ipc = 10`. We also study the influence of network architectures participating compensator generation. According to the results provided in Table 3, performance consistently improves with the addition of more ensembled networks. This trend is particularly pronounced with dynamic labeling. For instance, the ensemble of four architectures enhances performance by 4.3% with static labeling and by 16.3% with dynamic labeling. To align with the multi-model-aided compensator generation, we also employ multiple networks for soft label generation. Table 6 in Appendix A.5 presents the model performance trained with soft labels generated by various architecture ensembles.

**Cross-Architecture Generalization.** Table 4 presents the performance evaluation of synthetic CIFAR-100 dataset across different architectures, trained from scratch. When `ipc=10`, our INFER+Dyn and INFER methods outperform SRe2L across all architectures. For instance, INFER+Dyn achieves an accuracy of 52.3% on ResNet50, significantly higher than the 22.4% achieved by SRe2L. For `ipc=50`, the performance advantage of INFER+Dyn and INFER remains evident. INFER+Dyn reaches an accuracy of 70.0% on ResNet50, far surpassing SRe2L. Our INFER shows a well generalization abilities across different architectures. The cross-architecture results on ImageNet, shown in Table 2, further confirm the effectiveness of our methods. Specifically, INFER+Dyn and INFER demonstrate superior performance compared to SRe2L and G-VBSM across various ResNet architectures on ImageNet-1k.

Table 4: Cross-architecture performance of distilled dataset. Here, the synthetic CIFAR-100 datasets are evaluated by training ResNet-50, ResNet-101 (He et al., 2016), MobileNetV2 (Sandler et al., 2018), EfficientNetB0 (Tan & Le, 2019), and ShuffleNetV2 (Ma et al., 2018) from scratch. The best performers in each setting are highlighted in **red**.

| Networks | ipc = 10 | | | ipc = 50 | | |
|---|---|---|---|---|---|---|
| | SRe2L | INFER | INFER+Dyn | SRe2L | INFER | INFER+Dyn |
| ResNet50 | 22.4 ±1.3 | 43.3 ±0.4 | **52.3** ±0.4 | 52.8 ±0.7 | 62.4 ±0.6 | **70.0** ±0.2 |
| MobileNetV2 | 16.1 ±0.5 | **46.7** ±0.1 | 43.3 ±0.7 | 43.2 ±0.2 | 64.1 ±0.2 | **67.2** ±0.2 |
| EfficientNetB0 | 11.1 ±0.3 | 32.5 ±0.3 | **34.9** ±0.8 | 24.9 ±1.7 | 55.0 ±0.5 | **62.7** ±0.4 |
| ShuffleNetV2 | 11.8 ±0.7 | **38.7** ±0.2 | 30.9 ±0.5 | 27.5 ±1.1 | 61.5 ±0.2 | **62.1** ±0.2 |

Figure 5: Visualizations of loss landscapes in pixel space on CIFAR-100 dataset. The optimal decision boundary is supposed to have a rapid change in cross-entropy loss at the edge, indicating a clear and distinctive decision boundary. **Left:** A distinctive decision boundary trained on the original dataset $\mathcal{T}$. **Middle:** A less distinctive decision boundary trained on the synthetic dataset of outstanding class-specific approach SRe2L. **Right:** An improved decision boundary trained on the synthetic dataset of INFER.

### 4.4 More Discussions with SOTA Method SRe2L

**Feature Duplication Study.** We verify our hypothesis that synthetic data instances tend to capture distinctive yet duplicated intra-class features under the traditional class-specific distillation paradigm. We measure feature duplication by averaging the cosine similarities between each pair of synthetic data instances within the same class. Our experimental results, as shown in Figure 4(a), support our hypothesis: the class-specific approach SRe2L exhibits higher feature duplication as the number of ipc increases. Conversely, our INFER, achieves improved feature uniqueness, more closely resembling natural datasets. This reduction in intra-class feature duplication significantly enhances the diversity of the distilled dataset, which, in turn, improves the training performance.

**Visualization on Decision Boundaries.** To verify our hypothesis regarding the "oversight of inter-class features" in the traditional class-specific distillation paradigm, we visualize the decision boundaries of ResNet-18 models trained with synthetic datasets generated by SRe2L and our INFER, respectively. We randomly select seven classes from the CIFAR-100 dataset and use the t-SNE approach for visualization. As illustrated in Figure 2, INFER forms thin and clear decision boundaries between classes, in contrast to the chaotic decision boundaries produced by the traditional approach. Additionally, we visualized the 3D loss landscape in pixel spaces of the decision boundaries in Figure 5, which further supports our hypothesis from a different perspective.

For further analysis, refer to Appendix A.6, where we provide additional insights into the performance improvements of INFER.

## 5 Conclusion

In this work, we rethink the current "one class per instance" paradigm in dataset distillation and identified its limitations, including inefficient utilization of the distillation budget and oversight of inter-class features. These issues arise as distillation techniques advance, leading to synthetic data that often captures duplicated class-specific features. To address these limitations, we introduce a novel paradigm INFER that employs a Universal Feature Compensator (UFC) for "one instance for all classes" distillation. Our experimental results demonstrate that INFER improves the efficiency and effectiveness of dataset distillation, achieving state-of-the-art results in several datasets, reducing resource requirements while maintaining high performance. Future work will focus on scaling INFER for extremely large dataset and exploring its application in various real-world scenarios.

ACKNOWLEDGEMENT

This research is supported by Jiawei Du's A\*STAR Career Development Fund (CDF) C233312004.

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
