## A  APPENDIX

### A.1  TRAINING WITH SYNTHETIC DATASET

---

**Algorithm 2** Training with synthetic dataset via Inter-class Feature Compensator (INFER)

---

**Require:** Synthetic dataset $\mathcal{S}$; A network $f_\theta$ with weights $\theta$; Mini-batch size $b$; Learning rate $\eta$; Parameter $\beta$ for MixUP.

1: Initialize $\tilde{\mathcal{S}} = \{\}$
2: **for** each $\{\mathcal{P}^k, \mathcal{U}^k, \mathcal{Y}^k\}$ in $\mathcal{S}$ **do**
3:     ▷ Construct integrated synthetic instance
4:     $\tilde{\mathcal{S}}^k = \{(\tilde{\boldsymbol{s}}_i, \tilde{\boldsymbol{y}}_i) \mid \tilde{\boldsymbol{s}}_i = \boldsymbol{x}_i + \boldsymbol{u}_j, \text{ for each } \boldsymbol{x}_i \in \mathcal{P}^k \text{ and each } \boldsymbol{u}_j \in \mathcal{U}^k, \tilde{\boldsymbol{y}}_i \in \mathcal{Y}^k\}$
5:     $\tilde{\mathcal{S}} = \tilde{\mathcal{S}} \cup \tilde{\mathcal{S}}^k$
6: **end for**
7: ▷ Start training network $f_\theta$
8: **for** $e = 1$ to $E$ **do**
9:     Randomly shuffle the synthetic dataset $\tilde{\mathcal{S}}$
10:     **for** each mini-batch $\{(\tilde{\boldsymbol{s}}_i^{(b)}, \tilde{\boldsymbol{y}}_i^{(b)})\}$ **do**
11:         ▷ Augmented synthetic dataset by MixUP without additional relabel
12:         $\{(\tilde{\boldsymbol{s}}_i^{(b)}, \tilde{\boldsymbol{y}}_i^{(b)})\} = \text{MIXUP}(\{(\tilde{\boldsymbol{s}}_i^{(b)}, \tilde{\boldsymbol{y}}_i^{(b)})\})$
13:         Compute loss function $\mathcal{L}(f_\theta; \tilde{\boldsymbol{s}}_i^{(b)}, \tilde{\boldsymbol{y}}_i^{(b)})$                          ▷ KL Loss
14:         Update the weights: $\theta \leftarrow \theta - \eta \nabla_\theta \mathcal{L}$
15:     **end for**
16: **end for**
**Ensure:** The network $f_\theta$ with converged weights $\theta$
17: **function** MIXUP($\{(\boldsymbol{x}_i^{(b)}, \boldsymbol{y}_i^{(b)})\}, \beta$)
18:     $\{(\boldsymbol{x}'^{(b)}_i, \boldsymbol{y}'^{(b)}_i)\} \leftarrow \text{shuffle}\big(\{(\boldsymbol{x}_i^{(b)}, \boldsymbol{y}_i^{(b)})\}\big)$                    ▷ Shuffle the batch of inputs and labels
19:     Sample $\lambda$ from Beta($\beta, \beta$) for the batch
20:     $\lambda' \leftarrow \max(\lambda, 1 - \lambda)$                                          ▷ Ensure symmetry
21:     **for** $i = 1$ to $b$ **do**
22:         $\tilde{\boldsymbol{x}}_i \leftarrow \lambda' \boldsymbol{x}_i + (1 - \lambda') \boldsymbol{x}'_i$
23:         $\tilde{\boldsymbol{y}}_i \leftarrow \lambda' \boldsymbol{y}_i + (1 - \lambda') \boldsymbol{y}'_i$                              ▷ Linear interpolation of labels
24:     **end for**
        **return** $\{(\tilde{\boldsymbol{x}}_i^{(b)}, \tilde{\boldsymbol{y}}_i^{(b)})\}$
25: **end function**

---

### A.2  MORE RELATED WORKS

The goal of dataset distillation is to create a condensed dataset that, despite its significantly reduced scale, maintains comparable performance to the original dataset. This concept was first introduced by Wang et al. (Wang et al., 2018) as a bi-level optimization problem. Building on this foundational work, recent advancements have broadened the range of techniques available for effectively and efficiently condensing representative knowledge into compact synthetic datasets. Techniques such as gradient matching (Zhao et al., 2021; Zhao & Bilen, 2021; Lee et al., 2022b; Shin et al., 2023) optimize synthetic data to emulate the weight parameter gradients of the original dataset, while trajectory matching (Cazenavette et al., 2022; Cui et al., 2023; Du et al., 2023a;b) aims to replicate gradient trajectories to tighten synthesis constraints, showcasing the focus on effectiveness. Additionally, factorized methods (Liu et al., 2022; Kim et al., 2022; Wei et al., 2023; Shin et al., 2024) use specialized decoders to generate highly informative images from condensed features, enhancing both the utility and efficiency of distilled datasets. Another strategy, distribution matching (Wang et al., 2022; Zhao & Bilen, 2023; Sajedi et al., 2023; Sun et al., 2024; Deng et al., 2024), optimizes synthetic data to align its feature distribution with that of the original dataset on a class-wise basis. Although its efficiency has significantly improved, the performance of this method may still lag behind those using gradient or trajectory matching. Despite these innovations, most methods continue to grapple with balancing final accuracy and computational efficiency, presenting challenges for their application to large-scale and real-world datasets.

Table 5: Training recipes of CIFAR-10/100, Tiny-ImageNet, and ImageNet-1k.

| | | Optimizer | Learning Rate | Batch Size | Epoch/Iteration | Augmentation | Architectures |
|---|---|---|---|---|---|---|---|
| **CIFAR-10/100** | Syn | Adam $\{\beta_1, \beta_2\} = \{0.5, 0.9\}$ | 0.25 cosine decay | 100 | Iteration:1000 | - | {ResNet18, MobileNetv2, EfficientNetB0, ShuffleNetV2} |
| | Val | AdamW weight decay = 0.01 | 0.001 cosine decay | 64 | Epoch:400 | RandomCrop RandomHorizontalFlip MixUp | {ResNet18, MobileNetv2, EfficientNetB0, ShuffleNetV2} |
| | Val+dyn | AdamW weight decay = 0.01 | 0.001 cosine decay | 64 | Epoch:80 | RandomCrop RandomHorizontalFlip MixUp | {ResNet18, MobileNetv2, EfficientNetB0, ShuffleNetV2} |
| **Tiny-ImageNet** | Syn | Adam $\{\beta_1, \beta_2\} = \{0.5, 0.9\}$ | 0.1 cosine decay | 200 | Iteration:2000 | RandomResizedCrop RandomHorizontalFlip | {ResNet18, MobileNetv2, EfficientNetB0, ShuffleNetV2} |
| | Val | SGD weight decay = 0.9 | 0.2 cosine decay | 64 | Epoch:200 | RandomResizedCrop RandomHorizontalFlip MixUp | {ResNet18, MobileNetv2, EfficientNetB0, ShuffleNetV2} |
| | Val+dyn | SGD weight decay = 0.9 | 0.2 cosine decay | 64 | Epoch:50 | RandomResizedCrop RandomHorizontalFlip MixUp | {ResNet18, MobileNetv2, EfficientNetB0, ShuffleNetV2} |
| **ImageNet-1k** | Syn | Adam $\{\beta_1, \beta_2\} = \{0.5, 0.9\}$ | 0.25 cosine decay | 1000 | Iteration:2000 | RandomResizedCrop RandomHorizontalFlip | {ResNet18, MobileNetv2, EfficientNetB0} |
| | Val | AdamW weight decay = 0.01 | 0.001 cosine decay | 32 | Epoch:300 | RandomResizedCrop RandomHorizontalFlip MixUp | {ResNet18, MobileNetv2, EfficientNetB0} |
| | Val+dyn | AdamW weight decay = 0.01 | 0.001 cosine decay | 32 | Epoch:75 | RandomResizedCrop RandomHorizontalFlip MixUp | {ResNet18, MobileNetv2, EfficientNetB0} |

To further enhance dataset distillation for large datasets like ImageNet-1k (Deng et al., 2009), researchers have developed various innovative strategies to overcome inherent challenges. Cui et al. (Cui et al., 2023) introduced unrolled gradient computation with constant memory usage to manage computational demands efficiently. Following this, Yin et al. (Yin et al., 2024) developed the SRe2L framework, which decouples the bi-level optimization of models and synthetic data during training to accommodate varying dataset scales. Recognized for its excellent performance and adaptability, this framework has spurred further research. Further, Yin et al. proposed the Curriculum Data Augmentation (CDA) (Yin & Shen, 2023), a method that enhances accuracy without substantial increases in computational costs. Based on SRe2L, Zhou et al. (Zhou et al., 2024) developed SC-DD, a Self-supervised Compression framework for dataset distillation that enhances the compression and recovery of diverse information, leveraging the potential of large pretrained models. Additionally, Xuel et al. (Xue et al., 2024) focused on improving the robustness of distilled datasets by incorporating regularization during the squeezing stage of the SRe2L process. In Generalized Various Backbone and Statistical Matching (G-VBSM) (Shao et al., 2024), Shao et al. introduced a "local-match-global" matching technique based on SRe2L, which yields more precise and effective results, producing a synthetic dataset with richer information and enhanced generalization capabilities. Most recently, the Curriculum Dataset Distillation (CUDD) (Ma et al., 2024) was introduced, employing a strategic, curriculum-based approach to distillation that balances scalability and efficiency. This framework systematically distills synthetic images, progressing from simpler to more complex tasks.

## A.3 TRAINING RECIPES

The hyperparameter settings for the experimental datasets CIFAR-10/100, Tiny-ImageNet, and ImageNet-1k are listed in Table 5.

## A.4 VISUALIZATION OF GENERATED COMPENSATOR

To offer a clearer understanding of how the compensator enhances distillation outcomes, we visualize the compensators in Figure 7 and Figure 6. These visualizations reveal varying compensator patterns across different models and initialization instances.

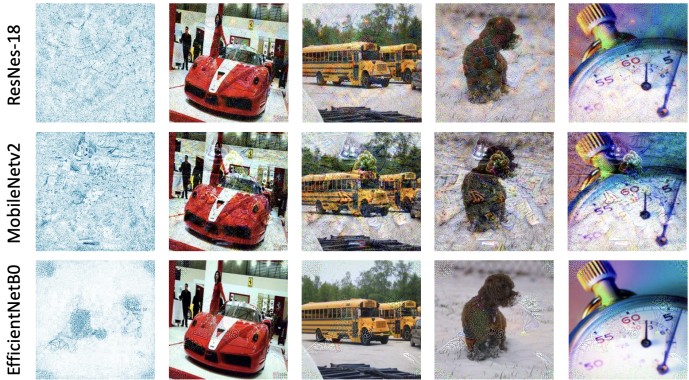

Figure 6: Visualizations of generated inter-class compensators of different models.

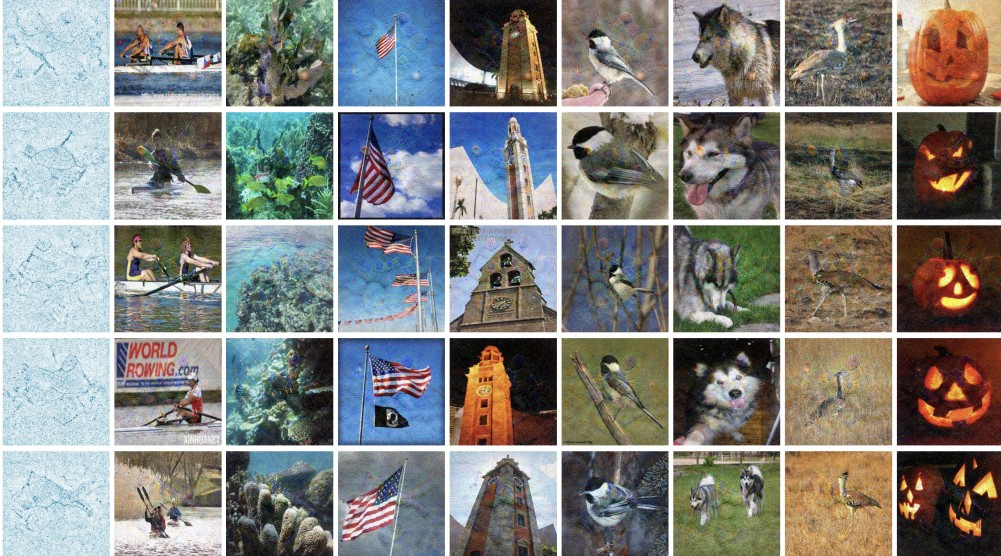

Figure 7: Visualizations of generated inter-class compensators using different initialization instances.

### A.5 ENSEMBLE OF ARCHITECTURES FOR SOFT LABEL GENERATION

### A.6 MORE DISCUSSIONS WITH SOTA METHOD SRE2L

**Generalization to Varying Sample Difficulties.** As illustrated in Figure 8, the x-axis shows the cross-entropy loss from a pretrained model, indicating sample difficulty. The graph reveals that IN-FER consistently outperforms SRe2L across all levels of sample difficulty, from easy to challenging. This demonstrates that INFER's performance improvements are comprehensive, providing a robust enhancement regardless of sample complexity.

**Adaptability to Static Labeling.** Figure 9 shows the Kullback-Leibler divergence (KLD) between dynamic and static labels. A smaller KLD indicates that our method adapts better to static labeling, making high compression rates feasible. In contrast, SRe2L relies heavily on dynamic labeling, which is more memory-intensive. This explains why our method achieves high compression rates while maintaining satisfactory performance.

Table 6: Ensemble of architectures for soft label generation. "R", "M", "E", and "S" represent ResNet18 (He et al., 2016), MobileNetv2 (Sandler et al., 2018), EfficientNetB0 (Tan & Le, 2019), and ShuffleNetV2 (Ma et al., 2018), respectively. ✔ indicates the network architectures participating in soft labels generation. ↑ denotes the performance gain contributed by the current ensembles compared with the baseline (only ResNet18). These experiments are conducted on CIFAR-100 dataset.

| R | M | E | S | ipc = 10 | | | | ipc = 50 | | | |
| | | | | INFER | ↑ | INFER+Dyn | ↑ | INFER | ↑ | INFER+Dyn | ↑ |
|---|---|---|---|---|---|---|---|---|---|---|---|
| ✔ | | | | 37.6 ±0.5 | + 0.0 | 46.4 ±0.2 | + 0.0 | 57.8 ±0.1 | + 0.0 | 69.0 ±0.1 | + 0.0 |
| ✔ | ✔ | | | 42.7 ±0.1 | + 5.1 | 53.2 ±1.0 | + 6.8 | 61.3 ±0.3 | + 3.5 | 70.2 ±0.3 | + 1.2 |
| ✔ | ✔ | ✔ | | 44.1 ±0.5 | + 6.5 | 53.0 ±0.6 | + 6.6 | 61.8 ±0.2 | + 4.0 | 69.0 ±0.1 | + 0.0 |
| ✔ | ✔ | ✔ | ✔ | 45.2 ±0.4 | + 7.6 | 53.4 ±0.6 | + 7.0 | 62.8 ±0.4 | + 5.0 | 68.9 ±0.1 | - 0.1 |

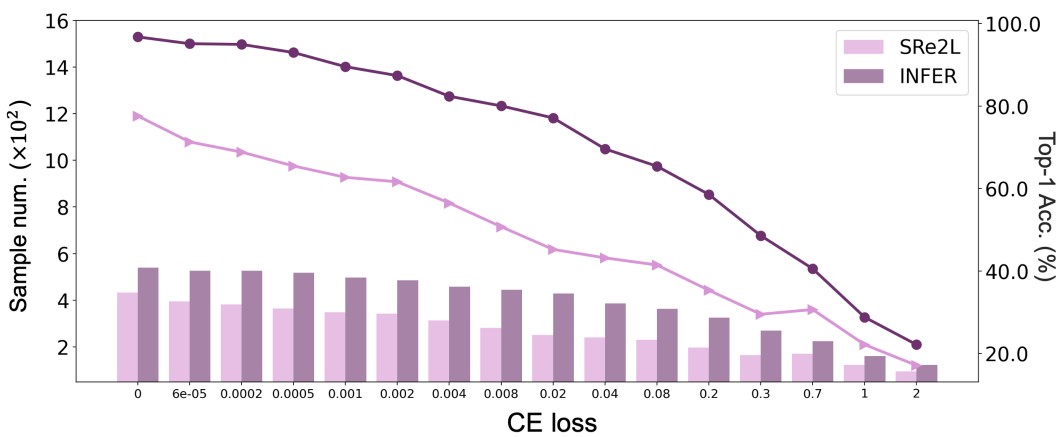

Figure 8: Performance on the validation set. The bars represent the number of samples correctly classified by SRe2L and our INFER, in each CE loss interval.

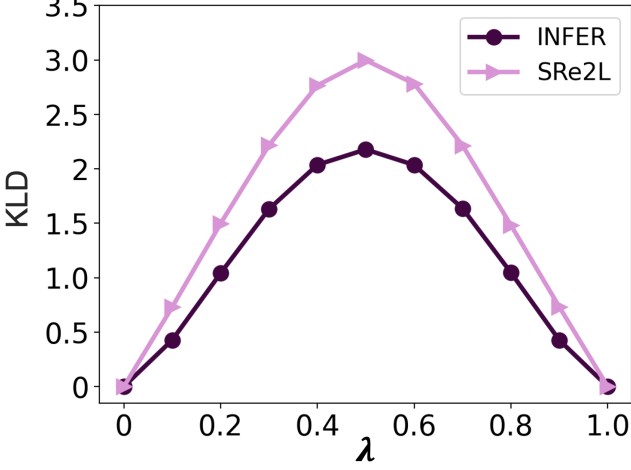

Figure 9: Kullback-Leibler divergence (KLD) between dynamic labels and static labels.