# OpenReview forum: "Breaking Class Barriers: Efficient Dataset Distillation via Inter-Class Feature Compensator"
_ICLR.cc/2025/Conference — ICLR 2025 Poster_

### Official Review · Reviewer_aHjp · 2024-10-29

**Soundness:** 3
**Presentation:** 3
**Contribution:** 3
**Rating:** 8
**Confidence:** 3

**Summary:**

The authors highlight that the “one label per instance” paradigm reduces intra-class diversity and amplifies inter-class differences within the distilled dataset. These limitations hinder the efficient use of the distillation budget and lead to poorly defined decision boundaries. To overcome these challenges, the authors propose a novel “one instance for all classes” approach, incorporating a Universal Feature Compensator (UFC). This method enhances class overlaps and ambiguities, improving representation quality and achieving state-of-the-art performance across a variety of datasets and architectures.

**Strengths:**

- The analysis of the two limitations in class-specific dataset distillation is insightful.
- The proposed Inter-class Feature Compensator is novel and improves the utilization of the distillation budget by leveraging inter-class features.
- To assess the size of the final distilled dataset, the authors introduce a compression ratio, in addition to the commonly used IPC metric, providing a fairer comparison with methods that utilize soft labels.
- The paper is well-written and well-structured.
- It offers extensive validation experiments across multiple datasets and architectures, reinforcing the reliability of the proposed approach.

**Weaknesses:**

- [1] explores inter-class and inter-class relations in dataset distillation, introducing a class centralization constraint to tighten within-class clustering and improve class discrimination. While both methods report significant performance improvements, the conclusions appear to contrast with those of this paper. Could the authors elaborate on this discrepancy?
- [2] also employs multiple backbone architectures for dataset distillation. It would be helpful if the authors discuss how their approach aligns with or differs from this.
- Since the optimizable parameter is not the entire distillation budget but only the M compensators, will this speed up the distillation or synthesis phase?
- How did the authors accurately calculate the compression ratio in Table 2? Additionally, how did the author estimate the size of soft labels?
- Does the compensator consume the IPC budget in the experiments? If yes, according to the design of compensator, the IPC values should not be integers.
- Including comparisons with recent state-of-the-art dataset distillation methods extended from SRe2L [3, 4] would provide a more comprehensive evaluation.
- There are some typos; for example, in line 401, “Our INFER also employ the soft labels” should be corrected to “Our INFER also employs the soft labels.”
[1] Exploiting Inter-sample and Inter-feature Relations in Dataset Distillation. CVPR 2024.
[2] Generalized Large-Scale Data Condensation via Various Backbone and Statistical Matching. CVPR 2024
[3] Dataset Distillation in Large Data Era
[4] On the Diversity and Realism of Distilled Dataset: An Efficient Dataset Distillation Paradigm. CVPR 2024

**Questions:**

please see the weakness part.

---

> ### Author Response · Authors · 2024-11-21
>
> Thank you for your insightful comments! We answer your questions as below.
>
> **Q1**(Weakness 1 & 2): discussion of related works.
>
> - inter-class and inter-class relations
>
> IID [1] introduces a class centralization constraint aimed at enhancing class discrimination by clustering samples more closely within classes. In contrast, our proposed INFER focuses on enhancing cross-class features to mitigate feature duplication. Both approaches are conceptually sound and cater to distinct use cases effectively.
> Class-centralized method, IID, excel on simpler datasets and with smaller IPC settings, where preserving the most representative features of each class is sufficient to maintain performance. However, as dataset complexity increases or the IPC grows, these representative features become inadequate to fully capture the dataset distribution. At this point, the advantages of our method, INFER, become evident. This is clearly demonstrated in following experimental results. Besides, IID not only centralizes features but also incorporates a covariance matching constraint to enhance feature diversity. This concept aligns with our approach, as both methods aim to reduce feature duplication and better approximate the real data distribution.
> - CIFAR-10:
>
> |Methods |  $\texttt{ipc} = 10$ | $\texttt{ipc} = 50$ |
> | -------- | -------- | -------- |
> | IID [1]    |  **59.9**  | 69.0     |
> | INFER     | 38.67     |  60.25    |
> | INFER+dyn     | 37.15      | **77.99**     |
>
> - CIFAR-100:
>
> |Methods |  $\texttt{ipc} = 10$ | $\texttt{ipc} = 50$ |
> | -------- | -------- | -------- |
> | IID [1]    | 45.7   | 51.3     |
> | INFER     | **49.59**     | **59.2**     |**60.15**|
> | INFER+dyn     | 48.37     | 57.53     | 58.97|
>
> - multiple backbone architectures for dataset distillation
>
> In the proposed INFER framework, we utilize multiple backbone architectures, each individually responsible for training the one compensator. The compensators learned by each architecture are illustrated in Appendix A.4, Figure 6, and the ablation studies examine each architecture’s contribution to the final performance are provided in Tables 4 and 6.
> Although G-VBSM [2] also employs diverse backbone architectures, it adopts a random sampling strategy, resulting in a synthesized dataset that is a mixture of outputs from all architectures. we evaluated the feasibility of this random sampling approach by incorporating it into our INFER framework.
>
> - CIFAR-100
>
> |  | $\texttt{ipc} = 10$ |$\texttt{ipc} = 50$ |
> | -------- | -------- | -------- |
> | INFER     | **50.2**    | **65.1**     |
> | INFER (random)   | 50.13     |64.37|
> | INFER +dyn     | **53.4**     | **68.9**     |
> | INFER (random) +dyn     | 53.05     |   67.68   |
>
> We observe that, given the same number of compensators, the random sampling method performs worse compared to the model-by-model strategy. This is because compensators optimized using randomly sampled architectures tend to converge toward a single optimal solution, thereby diminishing the diverse and complementary information provided by different architectures.
>
> **Q2**(Weakness 3): speed up distillation process.
>
> Thank you for the comment. Here, we provide a comparison of both computational cost and peak memory usage against existing methods following the defination in SRe2L [2].
>
> - Tiny-ImageNet: Time Cost represents the consumption (ms) when generating one image with one iteration update on synthetic data. Peak value of GPU memory usage is measured or converted with a batch size of 200.
>
> |Methods |  Condensed Arch |  Time Cost (ms)  |Peak GPU Memory Usage (GB)|
> | -------- | -------- | -------- |-------- |
> | DM [3]     | ConvNet     | 18.11     |10.7|
> | MTT [4]     | ConvNet     | 12.64     |48.9|
> | SRe2L [2]    | ResNet-18     | 0.75     |7.6|
> | INFER     | multi archs     |  0.21~0.71  | 7.6~9.1|
>
> As our distillation process involves multiple architectures, the time cost and GPU memory usage vary accordingly. While the number of optimizable parameters is reduced, capturing inter-class features requires the participation of original images in both forward and backward propagation. Consequently, the generation speed and memory requirements are similar to those of SRe2L.

---

> > ### Author Response · Authors · 2024-11-21
> >
> > **Q3**(Weakness 4): compression ratio calculation.
> >
> > The compression ratio (CR) is defined as the ratio of the synthetic dataset size to the original dataset size, calculated as  $\text{CR} = \frac{\text{Synthetic Dataset Size}}{\text{Original Dataset Size}}$  [5]. Below, we use $\texttt{ipc} = 50$ as an example to illustrate the calculation process. The original ImageNet-1K dataset occupies 138 GB of memory.
> >
> > Random: $50 \times 1000 /1281167 = 3.9\\%$
> >
> > SRe2L & G-VBSM & INFER+dyn: $([images:3.9 \\%\times 138G]+[labels:25.9G])/138G = 22.67\\%$
> >
> > INFER:$([images:3.9\\%\times138G]+[labels:0.1865G])/138G =4.04\\%$
> >
> > The label memory requirement for INFER is estimated based on a tensor with dimensions $50 \times 1 \times 1000 \times 1$, where 50 represents the $\texttt{ipc}$, 1000 denotes the number of classes in ImageNet-1K.
> >
> > **Q4**(Weakness 5): IPC budget allocation.
> >
> > After distillation, $P^k$ and $U^k$ $( k=1, \dots, K )$ are stored within the budget, where each $U^k$ consists of $M$ instances. Therefore, the total number of stored images is $K \times (C + M)$. To ensure fair comparisons, we calculate $K$ for each experimental IPC setting.
> > For instance, with $\texttt{ipc} = 10$ as indicated in Tables 1 and 2, $K$ is computed as:$K = 10 \times \frac{C}{C + M}$
> > For CIFAR-10 $(C = 10, M = 4)$, this gives: $K = \frac{10}{1.4} \approx 7$
> > As a result, the actual number of saved images per class is approximately 9.8. The fractional value arises because the $M \times K = 28$ compensators are shared across $C = 10$ classes. However, the size of the distilled dataset, calculated as $K \times (C + M) = 98$, remains an integer.
> >
> > **Q5**(Weakness 6): comparison with SOTAs.
> >
> > Thank you for your suggestion. To provide a more comprehensive evaluation, we have included comparisons with recent SOTA dataset distillation methods extended from SRe2L, such as RDED [3], CDA [4], and G-VBSM [2]. The results for various architectures and datasets are summarized below:
> >
> > - ResNet-18 ($\texttt{ipc} = 50$)
> >
> > |  | INFER | INFER+dyn | RDED [3] |CDA [4]|G-VBSM [2]|
> > | -------- | -------- | -------- | -------- |-------- |-------- |
> > | CIFAR-10     | 59.2    | 60.7     |**62.1**    |-|59.2|
> > | CIFAR-100     | 65.1     | **68.9**    |62.6     |64.4|65.0|
> > | Tiny-ImageNet      | 55.0     | 54.6     |**58.2**     |48.7|47.6|
> > | ImageNet-1K     | 54.3    | 55.6     |**56.5**     |53.5|51.8|
> >
> > - ResNet-50
> >
> > |  | INFER | INFER+dyn | CDA [4]|G-VBSM [2]
> > | -------- | -------- | -------- | -------- |-------- |
> > | Tiny-ImageNet ($\texttt{ipc} = 10$)     |**38.2** | 37.5|-|-|
> > | Tiny-ImageNet ($\texttt{ipc} = 50$)     |**55.52** |55.51|49.7|48.7|
> > | ImageNet-1K ($\texttt{ipc} = 10$)     | **39.3**|38.3|-|35.4|
> > | ImageNet-1K ($\texttt{ipc} = 50$)     | 62.5|**63.2**|61.3|58.7|
> >
> > - ResNet-101
> >
> > |  | INFER | INFER+dyn | RDED [3]| CDA [4]|G-VBSM [2]|
> > | -------- | -------- | -------- | -------- |-------- |-------- |
> > | Tiny-ImageNet ($\texttt{ipc} = 10$)    |**40.22** | 29.38|22.9|-|-|
> > | Tiny-ImageNet ($\texttt{ipc} = 50$)    | **56.13**|55.77|41.2|50.6|48.8
> > | ImageNet-1K ($\texttt{ipc} = 10$)    |37.0 |38.9|**48.3**|-|38.2|
> > | ImageNet-1K ($\texttt{ipc} = 50$)    |**63.0** |60.7|61.2|61.6|61.0|
> >
> > The results demonstrate that INFER is an effective dataset distillation method, consistently matching or surpassing SOTA methods across various datasets and settings.
> >
> > **Q6**(Weakness 7): writing typos.
> >
> > Thank you for pointing that out. We have carefully reviewed the submission and corrected the typos.
> >
> >
> > [1] Exploiting Inter-sample and Inter-feature Relations in Dataset Distillation. CVPR 2024.
> >
> > [2] Generalized large-scale data condensation via various backbone and statistical matching. CVPR 2024.
> >
> > [3] On the diversity and realism of distilled dataset: An efficient dataset distillation paradigm. CVPR 2024.
> >
> > [4] Dataset distillation in large data era. Arxiv 2024.
> >
> > [5]DC-BENCH: Dataset condensation benchmark. NeurIPS 2022

---

> > > ### Comment · Reviewer_aHjp · 2024-11-22
> > >
> > > Thank you for your thoroughly explanation and additional experiments. Most of my concerns have been solved. I will consider to increase my score.
> > >
> > > I still have two minor questions. Appreciate if you can give some explaination.
> > >
> > > First, I observed that the performance of INFER is inferior to RDED on ResNet-18. Could you provide some insights into the possible reasons for this discrepancy?
> > >
> > > Second, regarding the compression ratio calculation, I noticed that the distilled dataset (ipc = 50) downloaded from SRe2L is only 650 MB, whereas the calculation (3.9\% \times 138\,\text{GB} = 5.38\,\text{GB}) suggests a much larger size. Could you clarify the reason for this discrepancy?

---

> > > > ### Author Response · Authors · 2024-11-23
> > > >
> > > > Thank you for the prompt response!
> > > >
> > > > Q1: performance comparison with RDED.
> > > >
> > > > A1: The superior performance of RDED can be attributed to its use of dynamically generated soft labels, as emphasized in [1], which significantly boost performance. Unlike INFER, which relies on static soft labels, RDED updates its soft labels dynamically at each epoch. To isolate the impact of static labeling, we conducted a comparison of RDED’s performance using static labels under the same condition as INFER.
> > > >
> > > > - ImageNet-1K
> > > >
> > > > |  | RDED [1]| RDED with static labeling | INFER with static labeling|
> > > > | -------- | -------- | -------- |-------- |
> > > > | ipc=10| 42.0| 27.28 |**37.0**|
> > > > | ipc=50| 56.5| 44.4 |**54.3**|
> > > >
> > > > When evaluated under the same static labeling protocol, **RDED performs notably worse than INFER**.
> > > >
> > > > The non-optimization method introduced by RDED represents a patch-wise dataset pruning technique. Its performance can be further improved by integrating our optimization-based UFC approach.
> > > >
> > > > - ImageNet-1K (ipc=10):
> > > >
> > > > |  | RDED [1]| RDED + UFC |
> > > > | -------- | -------- | -------- |
> > > > | ipc=10| 42.0| **47.5** |
> > > >
> > > > As demonstrated by the above experimental results, the non-optimization method of RDED also overlooks inter-class features and can thus benefit from improvements introduced by our optimization-based UFC.
> > > >
> > > > Q2: distilled dataset size.
> > > >
> > > > A2:Thank you for pointing that out. Actually, the distilled dataset provided by SRe2L has already been resized to $224 \times 224$. For a fairer comparison with the original dataset, we should either calculate the proportion based on the number of images or compute the size of the resized version of the original dataset. Here, we opt for the first approach, as it is both simpler and more intuitive.
> > > >
> > > > [1] Qin, Tian, Zhiwei Deng, and David Alvarez-Melis. "A Label is Worth a Thousand Images in Dataset Distillation." in NeurIPS (2024).

---

> > > > > ### Comment · Reviewer_aHjp · 2024-11-23
> > > > >
> > > > > Thanks for your reply, my concerns have been solved and I have increased my score to 8.

---

> > > > > > ### Author Response · Authors · 2024-11-23
> > > > > >
> > > > > > Thank you for your time! We are so glad the responses addressed your concerns!

---

### Official Review · Reviewer_enyK · 2024-10-29

**Soundness:** 3
**Presentation:** 3
**Contribution:** 3
**Rating:** 6
**Confidence:** 4

**Summary:**

This paper presents a novel dataset distillation (DD) method, INFER (Inter-Class Feature Compensator), which explores the potential of leveraging inter-class features to significantly enhance distillation performance. In addition, it addresses a critical challenge faced by decoupled DD methods (e.g., SRe2L), where soft labels require nearly 30 times more storage than images. To overcome this limitation, the proposed method introduces a static labelling strategy that not only retains competitive performance but also dramatically reduces storage requirements, making it a more efficient and scalable solution.

**Strengths:**

a. The authors present an innovative analysis of existing class-specific dataset distillation methods. This work distinguishes itself from prior research by proposing the Universal Feature Compensator (UFC), which enhances inter-class feature integration, resulting in a more efficient dataset distillation framework. Relevant related works are appropriately cited. b. The submission is technically sound, with thorough evaluations and comparisons. c. The paper is clearly written and well-organized, making the content easy to follow and understand. d. This work addresses two key limitations of class-specific dataset distillation methods and achieves SOTA performance. Moreover, the proposed static labelling strategy reduces storage requirements by 99.3%.

**Weaknesses:**

a. The compensator is denoted by $\mu$ in Fig. 2, but by $u$ in Fig. 3. Maintaining consistent notation across the figures would improve clarity and help avoid potential confusion.
b. Some results fall short of SOTA benchmarks, and it would be helpful if the authors provided a discussion of potential reasons for this discrepancy.
c. The current evaluation only focuses on same-architecture distillation performance for ConvNet and ResNet18 on CIFAR and tiny-ImageNet. A broader cross-architecture evaluation would offer better insights into generalization.
d. While the static labelling approach effectively reduces storage requirements and, in some cases, even outperforms the dynamic method, the paper does not explore whether these benefits extend to other decoupled DD methods, such as SRe2L.

**Questions:**

e. I have a question regarding the IPC used in the experiments, which refers to the number of images per class. If the authors apply the augmentation approach described in Equation 2, the final number of distilled images per class becomes IPC \times \left(1 + \frac{M}{{class number}}\right) . This seems to differ slightly from the IPC used in other comparison methods. While this difference might have minimal impact on large datasets with many classes, it could be more influential for smaller datasets. Have the authors accounted for this adjustment to ensure a fair comparison?

---

> ### Author Response · Authors · 2024-11-21
>
> Thank you for your time and insightful feedback. We have provided detailed responses to all your questions below and hope they effectively resolve your concerns.
>
> **Q1**(Weakness 1): notion defination.
>
> Thank you for pointing that out. We have made the correction in the revised version (see Figure 2).
>
> **Q2**(Weakness 2): comparison with SOTAs.
>
> Thank you for the comment. As shown in Tables 1 and 2, INFER and INFER+dyn outperform other methods in most cases, except for the evaluation on CIFAR-10 with ConvNet under both $\texttt{ipc}=10$ and $\texttt{ipc}=50$. This is because, unlike other comparison methods, our methods do not involve the validation architecture (ConvNet) during the synthesis phase, thereby leading to cross-architecture generalization issues. To address this, we have included results where ConvNet is used during the distillation phase. Additionally, we incorporated a more recent SOTA method, DATM (ICLR 2024) [1], into the comparison.
>
> - CIFAR-10:
>
> |Methods |  $\texttt{ipc} = 10$ | $\texttt{ipc} = 50$ |
> | -------- | -------- | -------- |
> |Random|31.0|50.6|
> |DC|44.9|53.9|
> |DSA|52.1|60.6|
> |KIP|62.7|68.6|
> |RFAD|66.3|71.1|
> |MTT|65.4|71.6|
> |G-VBSM|46.5|54.3|
> | DATM [1]    |  **66.8**   | 76.1     |
> | IID [2]|59.9|69.0|-
> | SeqMatch   |  66.2   | 74.4    |
> | INFER     | 38.67     |  60.25    |
> | INFER+dyn     | 37.15      | **77.99**     |
>
> The performance of INFER+dyn shows significant improvement, surpassing other SOTA methods under the $\texttt{ipc}=50$ setting. However, at lower $\texttt{ipc}$ values, both INFER and INFER+dyn fail to provide notable benefits. This can be attributed to the fact that, under high compression ratios, preserving simple and easy-to-learn patterns, rather than cross-class compensatory features, is more effective in maintaining model performance. All the compared methods, being class-specific, naturally hold an advantage in such scenarios. Notably, INFER demonstrates its superiority as the distillation budget increases or when applied to more complex datasets, such as CIFAR-100 and Tiny-ImageNet.
>
> - CIFAR-100:
>
> |Methods |  $\texttt{ipc} = 10$ | $\texttt{ipc} = 50$ | $\texttt{ipc} = 100$
> | -------- | -------- | -------- |-------- |
> |Random|14.6|33.4|42.8|
> |DC|25.2|-|-|
> |DSA|32.3|42.8|-|
> |KIP|28.3|-|-|
> |RFAD|33.0|-|-|
> |MTT|39.7|47.7|49.2|
> |G-VBSM|38.7|45.7|-|
> | DATM [1]    |  47.2   | 55.0     |57.5|
> | IID [2]|45.7|51.3|-
> | SeqMatch    |  41.9   | 51.2    |-|
> | INFER     | **49.59**     | **59.2**     |**60.15**|
> | INFER+dyn     | 48.37     | 57.53     | 58.97|
>
>
> - Tiny-ImageNet:
>
> |Methods |  $\texttt{ipc} = 10$ | $\texttt{ipc} = 50$ |
> | -------- | -------- | -------- |
> |Random|5.0|15.0|
> |MTT|23.2|28.0|
> | DATM [1]    |  31.1  | 39.7    |
> | IID [2]|23.3|27.5|
> | SeqMatch   | 23.8  | -   |
> | INFER     |   33.03  |   38.93   |
> | INFER+dyn     | **35.49**     |  **39.83**  |
>
> \* Note that the results reported here involve ConvNet for both distillation and validation in these experiments, instead of ResNet-18 as described in the original submission.
>
>
> **Q3**(Weakness 3): cross-arch experiments.
>
> Thank you for the suggestion. We have incorporated additional cross-architecture experiments on CIFAR-10, CIFAR-100, and Tiny-ImageNet, utilizing networks such as MobileNetV2, EfficientNetB0, ShuffleNetV1, ResNet-50, and ResNet-101. These results demonstrate the robust cross-architecture generalization capability of our method.
>
> - CIFAR-10 ($\texttt{ipc} = 10$)
>
> | Models | MobileNetV2 | EfficientNetB0 | ShuffleNetV2 |
> | -------- | -------- | -------- | -------- |
> | SRe2L  |   17.71|11.47|13.42|
> | INFER     |**31.52**   | **29.14**     | **25.59**|
> | INFER +dyn    | 28.0     | 27.72     | 23.67|
>
> - CIFAR-10 ($\texttt{ipc} = 50$)
>
> | Models | MobileNetV2 | EfficientNetB0 | ShuffleNetV2 |
> | -------- | -------- | -------- | -------- |
> | SRe2L    |46.8|27.3|31.0
> | INFER     |**55.1**|**48.2**|**44.4**|
> | INFER +dyn    | 51.9|46.3|39.7|
>
> - CIFAR-100 ($\texttt{ipc} = 10$)
>
> | Models | ResNet-50|MobileNetV2 | EfficientNetB0 | ShuffleNetV2 |
> | -------- | -------- | -------- | -------- | -------- |
> | SRe2L     |22.4|16.1|11.1|11.8|
> | INFER     |49.3    | **50.8**     | **37.9**|**42.4**|
> | INFER +dyn    | **52.3**     | 43.3     | 34.9| 30.9|
>
> - CIFAR-100 ($\texttt{ipc} = 50$)
>
> | Models | ResNet-50|MobileNetV2 | EfficientNetB0 | ShuffleNetV2 |
> | -------- | -------- | -------- | -------- | -------- |
> | SRe2L      |52.8|43.2|24.9|27.5|
> | INFER     |65.6    | 66.9     | 60.5|**64.5**|
> | INFER +dyn    | **70.0**     | **67.2**    | **62.7**| 62.1|
>
> - Tiny-ImageNet ($\texttt{ipc} = 50$)
>
> | Models | ResNet-50|ResNet-101 |
> | -------- | -------- | --------
> | SRe2L  |42.2|42.5|
> | INFER    |**55.52**|**56.13**|
> | INFER +dyn |55.51| 55.77|

---

> > ### Author Response · Authors · 2024-11-21
> >
> > **Q4**(Weakness 3): effectiveness of static labeling.
> >
> > Thank you for raising this question. To address whether the benefits of the static labeling approach extend to other decoupled dataset distillation (DD) methods such as SRe2L, we conducted additional experiments, and the results are summarized in the table below:
> >
> > - CIFAR-100 (ResNet-18)
> >
> > | IPC | SRe2L | SRe2L with static labeling | INFER with static labeling
> > | -------- | -------- | -------- |-------- |
> > |10     | 31.6     | 13.7     |**50.2**
> > |50     | 49.5     | 31.36  |**65.1**
> >
> > The results indicate that applying static labeling to SRe2L significantly reduces its performance, suggesting that SRe2L relies heavily on dynamic feedback mechanisms during the validation process. In contrast, our method, INFER, achieves robust performance with static labeling, attributed to three key innovations outlined in lines 312–320. These innovations collectively enable INFER to deal with static labeling, delivering both efficiency and superior performance.
> >
> > **Q5**(Question 1): utilization of distillation budget.
> >
> > After distillation, $P^k$ and $U^k$ $( k=1, \dots, K )$ are stored within the budget, where each $U^k$ consists of $M$ instances. Therefore, the total number of stored images is $K \times (C + M)$. To ensure fair comparisons, we calculate $K$ for each experimental IPC setting.
> > For instance, with $\texttt{ipc} = 10$ as indicated in Tables 1 and 2, $K$ is computed as:$K = 10 \times \frac{C}{C + M}$
> > For CIFAR-10 $(C = 10, M = 4)$, this gives: $K = \frac{10}{1.4} \approx 7$
> > As a result, the actual number of saved images per class is approximately 9.8, ensuring a fair comparison.
> >
> > [1] Towards lossless dataset distillation via difficulty-aligned trajectory matching. ICLR 2024.
> >
> > [2] Exploiting Inter-sample and Inter-feature Relations in Dataset Distillation. CVPR 2024.

---

> > > ### Comment · Reviewer_enyK · 2024-11-23
> > >
> > > Thank you for your detailed response. Your clarifications have addressed all of my concerns. I recommend incorporating the results into your final version.

---

> > > > ### Author Response · Authors · 2024-11-23
> > > >
> > > > Thank you for your helpful review! We are glad to have addressed your concerns. These results will be incorporated into the final version.

---

### Official Review · Reviewer_hymx · 2024-11-01

**Soundness:** 3
**Presentation:** 3
**Contribution:** 2
**Rating:** 5
**Confidence:** 4

**Summary:**

In general, dataset distillation methods employ a class-specific synthesis paradigm, each synthetic instance has one label. This work suggests two limitations that these "one label per instance" approaches do not utilize the budget efficiently and fails to adequately reflect inter-class features. To overcome these limitations, this work introduces a new approach, Inter-class Feature Compensator (INFER), for “one instance for all classes” framework by utilizing Universal Feature Compensator (UFC). Experimental results show the effectiveness of the proposed framework.

**Strengths:**

1. This manuscript is well written and easy to follow along with their reasoning.
2. The idea of UFC is simple and easy to adapt.
3. The motivations of this work, inefficient budget utilization and oversight of inter-class features, are straightforward and novel. The proposed method aligns with these motivations.

**Weaknesses:**

1. The calculation of utilized budget of INFER is unclear. Some components in INFER seem to be consist of several vectors which have same size with original natural instance. However, in the manuscript, there is explicit description of utilized budget of INFER. Therefore, it is essential to clearly specify how the budget used by INFER is calculated and how it is set to ensure a fair comparison with prior studies.

2. There is insufficient baseline used for performance comparison. Although many dataset distillation methods are recently proposed to enhance the performance and budget efficiency, this work only compares INFER to some (possibly outdated) studies. Additionally, comparing INFER’s performance with recent methods raised doubts about its effectiveness. For example, DATM [1], which also address the feature redundancy as the manuscript said (Line 238-240), achieves higher performance than INFER in many settings: 47.2% (IPC=10), 57.5% (IPC=100) for CIFAR-100 and 31.1% (IPC=10). 39.7% (IPC=50) for Tiny-ImageNet.

3. One of the component in INFER, $P^k$, is consist of real instances. Storing real instances can be a drawback in tasks where privacy issues are crucial (e.g., continual learning). Continual learning is one of the primary applications of dataset distillation.

[1] Towards lossless dataset distillation via difficulty-aligned trajectory matching

**Questions:**

1. How to calculate the utilized budget of INFER? According to Algorithm 1, it seems that $P^k$, $U^k$, and $Y^k$ are stored in budget after training. $P^k$ consists of instances equal to the number of classes $C$ (Line 263), and $U^k$ consists of $M$ instances (Line 269). Since there are a total of $K=\text{IPC}$ (Line 261) sets of $P^k$ and $U^k$, the total number of instances in $P$ and $U$ amounts to $\text{IPC} \times (C + M)$, which already exceeds the budget of $\text{IPC}×C$ used by conventional methods. If my understanding is incorrect, please clarify the budget utilized by INFER.

2. Is there a reason for learning $U^k$ independently for each $k$? Learning $U^k$ independently for each $k$ aims to capture the inter-class features of $P^k$. This approach seems to capture only local inter-class features, which may be suboptimal. Introducing a single $U$ could potentially be beneficial for obtaining global inter-class features. Are there experimental results using a single $U$?

3. In Algorithm 1, all $u_j \in U^k$ are initialized with zero vector and independently optimized with same ingredient. This suggests a risk that all instances of $u_j$ may converge to the same optimal value. I am curious about how different values of the optimized $u_j$ actually are. Additionally, I am interested in the rationale for initializing with a zero vector rather than a random noise vector, as well as the resulting performance when using the latter.

4. In general, data factorization methods generate more instance than IPC under same budget. Given this characteristic, feature duplication may present a greater limitation in data factorization methods. Does feature duplication occur in data factorization as well? If it does, is there an improvement in performance when combining INFER with data factorization methodologies?

---

> ### Author Response · Authors · 2024-11-21
>
> Thank you for your time and constructive comments. We have responded to all your questions as follows and hope these address your concerns.
>
> **Q1**(Weakness 1 & Question 1): calculation of distillation budget
>
> **A1:** Thank you for bringing this to our attention. We must apologize for the incorrect statement regarding $K = \texttt{ipc}$ in line 261. As clarified in line 407, ‘Lastly, we also average the number of UFCs into ipc for fair comparison,’ the correct computation for $K$ is $K = \lfloor \texttt{ipc} \times \frac{C}{C + M} \rfloor$. Specifically, we use $K = 7, 35$ for CIFAR-10 with $\texttt{ipc} = 10, 50$; $K = 9, 48$ for CIFAR-100 with $\texttt{ipc} = 10, 50$; and $K = 9, 49$ for Tiny-ImageNet and ImageNet-1k with $\texttt{ipc} = 10, 50$.
>
> **Q2** (Weakness 2):comparison with SOTA method DATM [1]
>
> **A2:** Thank you for the suggestion. Several works aim to enhance the efficiency of budget utilization. For instance, DATM (ICLR 2024) [1], SeqMatch (NeurIPS 2023) [2], and PAD (Arxiv 2024) [3] have proposed various strategies to reallocate distillation budgets based on the difficulty of learning patterns.
> Below, we provide an experimental comparison on the CIFAR and Tiny-ImageNet datasets. To ensure fairness, ConvNet is involved in distillation and used for validation in these experiments, instead of ResNet-18 (as reported in the original submission).
> - CIFAR-10:
>
> |Methods |  ipc=10 | ipc=50 |
> | -------- | -------- | -------- |
> | DATM [1]    |  66.8   | 76.1     |
> | SeqMatch [2]    |  66.2   | 74.4    |
> | PAD [3]    |  **67.4**   |77.0    |
> | INFER     | 38.67     |  60.25    |
> | INFER+dyn     | 37.15      | **77.99**     |
>
> - CIFAR-100:
>
> |Methods |  ipc=10 | ipc=50 | ipc = 100
> | -------- | -------- | -------- |-------- |
> | DATM [1]    |  47.2   | 55.0     |57.5|
> | SeqMatch [2]    |  41.9   | 51.2    |-|
> | PAD [3]    |  47.8   |55.9   |58.5|
> | INFER     | **49.59**     | **59.2**     |**60.15**|
> | INFER+dyn     | 48.37     | 57.53     | 58.97|
>
>
> - Tiny-ImageNet:
>
> |Methods |  ipc=10 | ipc=50 |
> | -------- | -------- | -------- |
> | DATM [1]    |  31.1  | 39.7    |
> | SeqMatch [2]    | 23.8  | -   |
> | PAD [3]    |  32.3   |41.6   |
> | INFER     |   33.03  |   38.93   |
> | INFER+dyn     | **35.49**     |  **41.83** |
>
> The proposed INFER demonstrates better or comparable performance on these benchmarks. At smaller ipc, these class-specific distillation methods retain an advantage. This can be attributed to the fact that, under high compression ratios, retaining simple and easy-to-learn patterns rather than cross-class compensatory features effectively sustains model performance. Notably, INFER will demonstrate its superiority as the distillation budget increases or when applied to more complex datasets, such as CIFAR-100 and Tiny-ImageNet.
>
> **Q3** (Weakness 3): privacy-related question
>
> **A3:** There are two key reasons for including real instances. First, similar to the SOTA method RDED [4], real instances sampled from the original distribution enhance the realism and diversity of the distilled dataset, mitigating feature duplication and overfitting to specific distillation architecture.
> Second, while privacy is important, our primary focus is on efficiency. Real samples are essential for achieving an aggressive compression ratio through static labeling, as natural instances inherently follow the linear interpolation of labels.
> This is further supported by applying static labeling to SRe2L [5], a method that does not utilize real samples. The observed significant performance degradation highlights the critical role of real instances in improving efficiency. Moreover, the inferior performance of SRe2L to INFER also reinforces the importance of realism and diversity provided by real instances, as outlined in the first point.
>
> - CIFAR-100 (ResNet-18)
>
> | $\texttt{ipc}$ | SRe2L [5] | SRe2L with static labeling | INFER with static labeling
> | -------- | -------- | -------- |-------- |
> |10     | 31.6     | 13.7     |**50.2**
> |50     | 49.5     |    31.36  |**65.1**
>
> **Q4** (Question 2): local and global inter-class feature
>
> **A4:** Thank you for the question. We actually considered the so-called "global" UFCs but decided to deprecate them during the examination of our method. Below, we first present the comparison experiments between the "local" and "global" UFCs.
>
> - CIFAR-10 (ipc=10)
>
> |  | static labeling | dynamic labeling |
> | -------- | -------- | -------- |
> | "local" UFCs     | **33.5**     | **30.7**     |
> | "global" UFCs     | 31.7     | 27.7     |
>
> We elaborate on the reasons for deprecating the global UFCs as follows:
> * The performance drop is the primary reason to adopt "local" UFCs as shown in the above table.
> * Optimizing UFCs across the entire set $\mathcal{P}$ would incur extensive memory consumption, requiring $K$ times the memory of "global" UFCs.
> * The entire set $\mathcal{P}$ contains many instances belonging to the same class, causing the "global" UFCs to also capture intra-class features.

---

> > ### Author Response · Authors · 2024-11-21
> >
> > **Q5** (Question 3): UFC initialization
> >
> > **A5:** This is another intersting thinking we have considered when designing the algorithm to optimize UFCs. We concluded that zero initialization or random initialization **has no significant effect on the final performance**, as demonstrated below:
> > - CIFAR-100 ($\texttt{ipc}=50$)
> >
> > | init | INFER | INFER+dyn |
> > | -------- | -------- | -------- |
> > | random noise     |  65.67   | 68.68     |
> > | zero    | 65.1     | 68.9      |
> >
> > >all instances of $u_j$ may converge to the same optimal value
> >
> > The converged values of $u_j$ will vary across $\mathcal{P}^k$ because, mathematically, $u_j$ represents the shared values that minimize the overall loss of $\mathcal{P}^k$ when integrated into each element of $\mathcal{P}^k$.
> >
> > > how different values of the optimized $u_j$ actually are
> >
> > We have visualized the optimized $u_j$ in Appendix A4 Figure 6, which demonstrates their distinct patterns.
> >
> > >the rationale for initializing with a zero vector rather than a random noise vector
> >
> > We adopt zero initialization to avoid potential bias from random initialization. Since $u_j$ represents shared values that minimize the overall loss of $\mathcal{P}^k$, zero initialization has no impact on the loss before optimization begins. In contrast, random initialization may unevenly affect instance losses in $\mathcal{P}^k$. Additionally, as the solution for $u_j$ is not unique, the bias may not be fully removed after optimization.
> >
> > **Q6** (Question 4): combination with factorization-based methods
> >
> > **A6:** Thank you for bringing up this interesting question. We appreciate your input and will explore it further. Factorization-based methods, such as FreD [6], IDC [7], and HaBa [8], utilize transformations or auxiliary networks to generate more instances than IPC. Their performance improvement may stem from increased information capacity or reduced feature duplication. We plan to examine this by integrating our proposed approach with factorization methods. However, our verification experiments are still ongoing due to the extensive training time required by these methods.
> >
> > [1] Towards lossless dataset distillation via difficulty-aligned trajectory matching. ICLR 2024.
> >
> > [2] Sequential subset matching for dataset distillation. NeurIPS 2024.
> >
> > [3] Prioritize Alignment in Dataset Distillation." arXiv 2024.
> >
> > [4] On the diversity and realism of distilled dataset: An efficient dataset distillation paradigm. CVPR 2024.
> >
> > [5] Squeeze, recover and relabel: Dataset condensation at imagenet scale from a new perspective. NeurIPS 2024.
> >
> > [6] Frequency Domain-based Dataset Distillation, NeurIPS 2023.
> >
> > [7] Dataset condensation via efficient synthetic-data parameterization. ICML 2022.
> >
> > [8] Dataset distillation via factorization. NeurIPS 2022.

---

> ### Author Response · Authors · 2024-11-24
> **A warm reminder**
>
> Dear reviewer hymx,
>
> Thank you for your questions regarding the review of our paper. We understand that your schedule is very busy, but we wanted to kindly remind you that the discussion period is nearing its conclusion. You raised several questions during the first round of review, and we have carefully addressed each of them in our rebuttal.
>
> All other reviewers have responded to our rebuttal, and we sincerely hope you can review the responses, and we could help you address all your concerns about our submission.
>
> Thank you for your time and thoughtful consideration.
>
> Best The authors

---

> ### Comment · Reviewer_hymx · 2024-11-25
>
> I would like to thank the authors for your response and most of my concerns are resolved. However, I still have doubts about the effectiveness of the proposed idea under high compression ratios.
>
> The authors explained that the class-specific distillation method is more effective at preserving simple and easy-to-learn patterns under high compression ratios. If so, does this mean that the cross-class distillation method structurally fails to preserve such simple and easy-to-learn patterns effectively? If the cross-class distillation method is indeed ineffective under high compression ratios, I believe it has limitations as a universally applicable approach in the most challenging tasks (e.g., IPC = 1). I believe there should be a clear explanation and justification for this. Furthermore, I think Figure 1, which appears to suggest its effectiveness even under high compression ratios, needs appropriate revisions.
>
> Additionally, consistent statements regarding high compression ratios are necessary. I believe the authors used the justification of simple and easy-to-learn patterns to defend the significant performance gap with prior studies when considering CIFAR-10 and IPC=10. However, in Table 2, the proposed idea demonstrates higher performance than the baseline while using a lower budget (high compression ratio), which the authors emphasized. These contradictory experimental results make it difficult to clearly understand the effectiveness under high compression ratios.
>
> Nevertheless, I agree with defining the performance degradation of dataset distillation as IPC increases as a feature duplication problem and proposing an idea to mitigate it. Thus I decide to increase my score to 5, but I will not insist on rejection if all other reviewers champion acceptance.

---

> > ### Author Response · Authors · 2024-11-27
> > **Thanks for the comments and raising the score!**
> >
> > Dear reviewer hymx,
> >
> > Thank you for your prompt response! We deeply appreciate your rigorous review and proactive attitude toward improving the paper under review, both of which are crucial for the research community.
> >
> > We believe our disagreement primarily stems from the definition of a high compression ratio. In our view, the threshold for a high compression ratio lies between IPC=1 and IPC=5, depending on the specific dataset. Therefore, we agree with your suggestion that “consistent statements regarding high compression ratios are necessary,” and we will revise this aspect in our submission. However, we may retain Figure 1, as IPC=10 does not fall under the definition of “high compression ratio” for CIFAR-100, Tiny-ImageNet, and ImageNet-1k datasets.
> >
> > Additionally, we agree with your statement that “it has limitations as a universally applicable approach in the most challenging tasks (e.g., IPC = 1).” This is because we lack the distillation budget for UFC under the IPC = 1 task. Under such conditions, our INFER reduces to random selection, which is why we do not report results for IPC = 1 tasks.
> >
> > The motivation of our INFER is, to some extent, to achieve lossless distillation on ImageNet-1k datasets with a relatively lower compression ratio. We observed that recent works in dataset distillation, particularly data parameterization-based methods, exhibit remarkable performance enhancements at IPC=1. However, **these impressive improvements tend to diminish significantly as the IPC increases**. This phenomenon can be explained by the increase in condensed information driving the performance improvement at IPC=1, while feature duplication accounts for the diminished performance as IPC increases.
> >
> > Therefore, we believe the future direction of dataset distillation **should not focus solely on improved condensation in scenarios with limited distillation budgets** (particularly low IPC). Instead, the goal should be to **sustain performance enhancement as the distillation budget increases**, ultimately achieving lossless distillation on ImageNet-1k datasets. By pursuing this approach, dataset distillation can evolve into a practical technique, aligning with the developmental trajectory of model compression.
> >
> > Thank you once again for your insightful responses, which have inspired the further reflections outlined above. With the discussion period extended, we look forward to hearing more from you—not only regarding our submission but also your thoughts on the future direction of this field!
> >
> > Best regards
> >
> > The authors

---

### Official Review · Reviewer_RT9a · 2024-11-03

**Soundness:** 2
**Presentation:** 3
**Contribution:** 2
**Rating:** 3
**Confidence:** 4

**Summary:**

This paper first identifies two limitations of the current dataset distillation methods: inefficient utilization of the distillation budget and oversight of inter-class features. Based on this, this paper proposes to leverage a Universal Feature Compensator to enhance feature integration across classes.
Experiments demonstrate that the proposed INFER improves the storage efficiency and effectiveness of dataset distillation.

**Strengths:**

1. It analyses the limitations of the current optimization-based dataset distillation methods.
2. The proposed method can significantly reduce the storage.
3. The paper is well-written.

**Weaknesses:**

1. The authors claim that the existing dataset distillation methods focus on optimizing synthetic data exclusively for a pre-assigned one-hot label. However, RDED [1] introduces a non-optimization method that changes this paradigm. It is recommended that the authors analyze and compare their method with RDED.

2. In Table 1, lines 416-417, the authors claim that static labels perform better on small networks, while dynamic labels excel on large networks. However, in Table 2, static labels sometimes outperform dynamic labels on large networks and datasets, particularly when
IPC=50 using ResNet-101. Why is this the case?

3. Additionally, in Table 4, the performance comparison between static and dynamic labels is inconsistent. Sometimes static labels are superior, other times dynamic labels are. What factors determine the choice between static and dynamic labels in real-world applications?

4. Analyzing computational cost is crucial in dataset distillation. It would be beneficial to compare both computation cost and peak memory usage against existing methods.

[1] Sun, Peng, et al. "On the diversity and realism of distilled dataset: An efficient dataset distillation paradigm." Proceedings of the IEEE/CVF Conference on Computer Vision and Pattern Recognition. 2024.

**Questions:**

1. The authors claim that the existing dataset distillation methods focus on optimizing synthetic data exclusively for a pre-assigned one-hot label. However, RDED [1] introduces a non-optimization method that changes this paradigm. It is recommended that the authors analyze and compare their method with RDED.

2. In Table 1, lines 416-417, the authors claim that static labels perform better on small networks, while dynamic labels excel on large networks. However, in Table 2, static labels sometimes outperform dynamic labels on large networks and datasets, particularly when
IPC=50 using ResNet-101. Why is this the case?

3. Additionally, in Table 4, the performance comparison between static and dynamic labels is inconsistent. Sometimes static labels are superior, other times dynamic labels are. What factors determine the choice between static and dynamic labels in real-world applications?

4. Analyzing computational cost is crucial in dataset distillation. It would be beneficial to compare both computation cost and peak memory usage against existing methods.

[1] Sun, Peng, et al. "On the diversity and realism of distilled dataset: An efficient dataset distillation paradigm." Proceedings of the IEEE/CVF Conference on Computer Vision and Pattern Recognition. 2024.

---

> ### Author Response · Authors · 2024-11-21
>
> Thank you for your constructive comments! We give point-to-point replies to your questions in the following.
>
> **Q1**(Weakness 1 & Question 1): comparison with RDED
>
> **A1:** We compare the performance of our proposed INFER to RDED as you suggested as below:
> |  | INFER | RDED [1] |
> | --------  | -------- | -------- |
> | CIFAR-10 (ipc 50)     | 59.2         |**62.1**    |
> | CIFAR-100 (ipc 50)     | **65.1**         |62.6     |
> | Tiny-ImageNet (ipc 50)     | 55.0        |**58.2**     |
> | ImageNet-1K (ipc 50)     | 54.3        |**56.5**     |
>
> Our INFER outperforms RDED on CIFAR-100 but falls short on the other three datasets. This difference can be attributed to the use of **dynamically generated soft labels in RDED**, as highlighted in [5], which significantly enhance performance. While RDED employs dynamic soft labels updated each epoch, INFER relies on static soft labels. To isolate the effect of static labeling, we compare the performance of RDED using static labels under the same conditions as INFER:
>
> - ImageNet-1K
>
> |  | RDED [1]| RDED with static labeling | INFER with static labeling|
> | -------- | -------- | -------- |-------- |
> | ipc=10| 42.0| 27.28 |**37.0**|
> | ipc=50| 56.5| 44.4 |**54.3**|
>
> The performance of **RDED lags significantly behind INFER** when evaluated **under the same static labeling protocol**.
>
> The non-optimization method proposed by RDED is a type of patch-wise dataset pruning technique. The performance of RDED can be further enhanced by applying our optimization-based UFC to it.
> - ImageNet-1K (ipc=10):
>
> |  | RDED [1]| RDED + UFC |
> | -------- | -------- | -------- |
> | ipc=10| 42.0| **47.5** |
>
> As evidenced by the above experimental results, non-optimization method of RDED also overlooks the inter class features and can therefore be improved by our optimization-based UFC.
>
> **Q2**(Weakness 2 & Weakness 3 & Question 2 & Question 3): explaination of experimental results.
>
> **A2:** We appreciate you bringing this to our attention and apologize for the imprecise conclusion in our submission. We will revise our statement to: “Static labels suffice for simpler networks, but dynamic labels are observed to offer stronger supervision for some more complex architectures.”
>
> We performed a statistical analysis and demonstrate this tendency as below. In the following tables, a ticked cell indicates better performance, and the architectures are roughly arranged by complexity from simple to complex across the rows.
>
> - CIFAR-100 (ipc=10)
>
> | Models | static label | dynamic label |
> | -------- | -------- | -------- |
> | ConvNet     | $\checkmark$     |      |
> | MobileNetV2     |  $\checkmark$     |      |
> | ShuffleNetV2     |  $\checkmark$     |      |
> | EfficientNetB0     |  $\checkmark$     |      |
> | ResNet-18     |      |   $\checkmark$    |
> | ResNet-50     |       |   $\checkmark$   |
>
> - CIFAR-100 (ipc=50)
>
> | Models | static label | dynamic label |
> | -------- | -------- | -------- |
> | ConvNet     | $\checkmark$     |      |
> | ShuffleNetV2     |    $\checkmark$   |      |
> | EfficientNetB0     |      |  $\checkmark$     |
> | ResNet-18     |      |   $\checkmark$    |
> | ResNet-50     |       |   $\checkmark$   |
>
> For CIFAR-100, regardless of whether ipc=10 or 50, dynamic labeling demonstrates superior performance on larger networks such as ResNet-18 and ResNet-50.
>
> - ImageNet-1K (ipc=10)
>
> | Models | static label | dynamic label |
> | -------- | -------- | -------- |
> | ResNet-18     |   $\checkmark$   |     |
> | ResNet-50     |    $\checkmark$   |     |
> | ResNet-101     |       |   $\checkmark$   |
>
>
> - ImageNet-1K (ipc=50)
>
> | Models | static label | dynamic label |
> | -------- | -------- | -------- |
> | ResNet-18     |     |    $\checkmark$   |
> | ResNet-50     |      |   $\checkmark$    |
> | ResNet-101     |   $\checkmark$     |     |
>
> For ImageNet-1K, the results remain consistent across architectures except for ResNet-101, which appears to be an outlier in these experiments.

---

> > ### Author Response · Authors · 2024-11-21
> >
> > **Q3**(Question 4): computation cost and peak memory usage
> >
> > **A3:** Thank you for the suggestion. Here, we provide a comparison of both computational cost and peak memory usage against existing methods following the definations in SRe2L [2].
> >
> > - Tiny-ImageNet: Time Cost represents the consumption (ms) when generating one image with one iteration update on synthetic data. Peak value of GPU memory usage is measured or converted with a batch size of 200.
> >
> > |Methods |  Condensed Arch |  Time Cost (ms)  |Peak GPU Memory Usage (GB)|
> > | -------- | -------- | -------- |-------- |
> > | DM [3]     | ConvNet     | 18.11     |10.7|
> > | MTT [4]     | ConvNet     | 12.64     |48.9|
> > | SRe2L [2]    | ResNet-18     | 0.75     |7.6|
> > | INFER     | multi archs     |  0.21~0.71  | 7.6~9.1|
> >
> > As multiple architectures are involved in our distillation process, the time cost and GPU memory usage vary accordingly. Overall, we maintain comparable generation speed and memory requirements to SRe2L.
> >
> > [1] Sun, Peng, et al. "On the diversity and realism of distilled dataset: An efficient dataset distillation paradigm." Proceedings of the IEEE/CVF Conference on Computer Vision and Pattern Recognition 2024.
> >
> > [2] Yin, Zeyuan, Eric Xing, and Zhiqiang Shen. "Squeeze, recover and relabel: Dataset condensation at imagenet scale from a new perspective." Advances in Neural Information Processing Systems 2024.
> >
> > [3] Cazenavette, George, et al. "Dataset distillation by matching training trajectories." Proceedings of the IEEE/CVF Conference on Computer Vision and Pattern Recognition. 2022.
> >
> > [4] Zhao, Bo, and Hakan Bilen. "Dataset condensation with distribution matching." Proceedings of the IEEE/CVF Winter Conference on Applications of Computer Vision. 2023.
> >
> > [5] Qin, Tian, Zhiwei Deng, and David Alvarez-Melis. "A Label is Worth a Thousand Images in Dataset Distillation." in NeurIPS (2024).

---

> ### Comment · Reviewer_RT9a · 2024-11-23
>
> Thank you for the detailed feedback. However, I still have some concerns:
>
> (1) The results presented about [Weakness 1 & Question 1] indicate that the proposed method underperforms RDED across most scenarios. The shortfall is attributed to static labels that are designed for label storage efficiency. However, prior research [1] has demonstrated that under substantial label space compression (e.g., 40× reduction on ImageNet-1K), their approach achieves superior performance compared to state-of-the-art methods, including RDED [2] and CDA [3]. This raises concerns regarding the novelty and competitiveness of the proposed method.
>
> (2) Furthermore, the observation that static labels outperform dynamic labels in some scenarios needs a deeper analysis. Could you provide a comprehensive explanation of the underlying factors driving this phenomenon?
>
> [1] Are Large-scale Soft Labels Necessary for Large-scale Dataset Distillation? NeurIPS, 2024.
>
> [2]  On the diversity and realism of distilled dataset: An efficient dataset distillation paradigm. CVPR, 2024.
>
> [3] Dataset distillation in large data era.

---

> ### Author Response · Authors · 2024-11-24
> **Thank you for your further feedback!**
>
> Thank you for your insightful response!
>
> **Q1 :**Performance Comparison with prior research [1]:
>
> **A1: Our method outperforms LPLD[1] significantly on the ImageNet-1k**, as indicated below:
>
> Table 1: Results of Resnet-18 on ImageNet-1k.
> |ResNet-18 |$40\times$ compressed labels RDED[2] |$40\times$ compressed labels CDA[3] | $40\times$ compressed labels LPLD[1]  |$300\times$ compressed labels Ours INFER|
> | --------  | -------- | -------- | -------- | -------- |
> | IPC 10 |24.0|13.2 |20.2 | **37.0** |
> | IPC 50 |44.3|38.0| 46.7 | **54.3** |
>
> Table 2: Results of Resnet-50, ResNet-101 on ImageNet-1k.
>
> |IPC=50 | $40\times$ compressed labels LPLD[1]  |$300\times$ compressed labels Ours INFER|
> | --------  | -------- | -------- |
> | ResNet-50 |42.3 | **62.5** |
> | ResNet-101 | 42.1 | **63.0** |
>
> All results of LPLD [1] are cited from their paper [1]. The results indicate:
>
> * Our proposed method, INFER (with static labels), achieves **significantly better performance than the previous state-of-the-art (SOTA) LPLD [1]** (with improvements ranging from 7.6% to 20.9% on ImageNet-1k).
> * More importantly, our INFER (with static labels) is **equivalent to $300\times$ compressed labels** under the same standard as stated in [1]. (This is because SRe2L, CDA [3], and RDED [2] use dynamic soft labels over 300 epochs, whereas INFER uses static labels in just 1 epoch, which is $300\times$ compressed labels.) This demonstrates that our INFER not only outperforms the previous SOTA LPLD [1] but does so with a much lower label budget, **highlighting the superior competitiveness of our method**.
>
> Additionally, LPLD [1] was released on ArXiv on **October 21, 2024**, and is motivated by the need to prune the soft labels of SRe2L. In contrast, our INFER, submitted on **October 1, 2024**, is driven by the goal of leveraging overlooked inter-class features to achieve significantly better performance than LPLD [1]. Therefore, we disagree with the claim that "This raises concerns regarding the novelty and competitiveness of the proposed method."
>
>
>
> **Q2 :** the observation that static labels outperform dynamic labels
>
> **A2:** We appreciate your keen observation and insightful question! This is indeed a very critical and interesting point that warrants further investigation. To address your query, we use the results from ImageNet-1k (ipc=50) as a case study.
>
> We observe that the "dynamic labels" approach outperforms "static labels" on both ResNet-18 (Dynamic: 55.6% vs. Static: 54.3%) and ResNet-50 (Dynamic: 63.4% vs. Static: 62.5%). However, it performs significantly worse than "static labels" on the largest dataset, ResNet-101 (Dynamic: 60.7% vs. Static: 63.0%). The **superior performance of "static labels" on ResNet-101** is attributed to the degradation of "dynamic labels" in this case. This degradation occurs because the "dynamic labels" approach relies on the technique of knowledge distillation.
>
>
> The synthetic data augmented by is equvalient to the generated data of data-free knowledge distillation, which transfers knowledge through the generated soft labels of the "teacher network" in every epoch to train the "student network". Consequently, the student network is expected to have a **simpler architecture** and exhibit **worse performance** compared to the teacher network.
>
>
>
> We list the number of parameters for each networks in experiments as below:
> |Architectures | Number of Paras  |Roles|
> | --------  | -------- | -------- |
> | MobileNetV2 |3.4M | Teacher |
> | EfficientNetB0 |5.3M | Teacher |
> | ResNet-18 |11.7M | Teacher/Student |
> | ResNet-50 |25.6M | Student |
> | ResNet-101 | 44.6M | Student |
>
> The results of "dynamic labels" on ResNet-101 degrade significantly because the teacher network, ResNet-18, is **much simpler than the student network**, ResNet-101. Such degration is also observed in LPLD[1] (ResNet-50 42.3% vs ResNet-101 42.1%) as indicated in Table 2 of our A1. However, our method, INFER, is not affected by the performance degradation caused by the network complexity gap. Unlike the "dynamic labels" approach, our INFER does not rely on Knowledge Distillation but instead adheres to the natural dataset principle of "one instance, one label." This highlights the great potential of INFER to achieve superior distillation performance.
>
>
>
> [1] Are Large-scale Soft Labels Necessary for Large-scale Dataset Distillation? NeurIPS, 2024
> [2] On the diversity and realism of distilled dataset: An efficient dataset distillation paradigm. CVPR, 2024.
> [3] Dataset distillation in large data era.

---

> > ### Comment · Reviewer_RT9a · 2024-11-25
> > **Fairness of the performance comparison.**
> >
> > Thank you for the detailed response. However, I still have concerns regarding the fairness of the comparative evaluation.
> >
> > This paper claims that the hyperparameter settings of SRe2L are adhered to, and the results for SRe2L, presented in Table 2, are directly adopted from [1] that proposes SRe2L. However, **a significant discrepancy exists in the batch size** used during the evaluation phase on distilled data: SRe2L employs a batch size of 1024, as detailed in Table 7 of [1], while this paper uses a batch size of 32 (as shown in Table 5). Such a deviation is non-trivial, given that **smaller batch sizes have been empirically shown to often result in improved performance**. This inconsistency undermines the fairness of the comparative analysis.
> >
> > Furthermore, for comparisons with LPLD[2], RDED[3], CDA[4], and, this paper directly copies the results reported in the original papers for these methods without verifying whether the comparisons are conducted under identical hyperparameter settings. For example, both LPLD and CDA utilize a batch size of 128, as reported in their original implementations, which contrasts with the batch size of 32 employed in this paper's experiments.
> >
> > These discrepancies in experimental settings, particularly with respect to batch size, significantly compromise the fairness and reliability of the comparisons.
> >
> > [1] Squeeze, Recover and Relabel: Dataset Condensation at ImageNet Scale From A New Perspective. NeurIPS, 2023
> >
> > [2] Are Large-scale Soft Labels Necessary for Large-scale Dataset Distillation? NeurIPS, 2024
> >
> > [3] On the diversity and realism of distilled dataset: An efficient dataset distillation paradigm. CVPR, 2024.
> >
> > [4] Dataset distillation in large data era.

---

> > > ### Author Response · Authors · 2024-11-27
> > > **We conduct more experiments as you suggested**
> > >
> > > Thank you for your suggestions regarding fairness in performance comparisons!
> > >
> > > As suggested, we reproduced CDA [1] and LDLP [2] results using their recommended batch size of 128. To ensure a fair comparison, we also used the same batch size of 128 to reproduce our INFER results.
> > >
> > > - **ImageNet-1K batch size = 128**
> > >
> > > | | CDA （$40 \times$ compressed labels）|LPLD [2]（$40 \times$ compressed labels）|INFER （$300 \times$ compressed labels） |
> > > |-------- |  -------- |-------- |-------- |
> > > | ipc = 10 |13.7  |20.0| **34.85**|
> > >
> > > The results still verify that our INFER achieves **14.85% performance improvement**  while using **significantly fewer soft labels** ($300\times$ compressed labels for INFER vs. $40\times$ for LPLD) compared to the previous state-of-the-art (SOTA) LPLD [1] on ImageNet-1k datasets with ipc=10 and batch size=128.
> > >
> > > We also verified your claim that a smaller batch size leads to performance improvements, as our INFER's performance degrades from 37.0% to 34.85% when using a larger batch size (from 32 to 128). However, the performance improvements is marginal, which is also indicated in the Table 6 of CDA [1].
> > >
> > > We first conducted experiments with IPC=10 to address your concerns, given the extensive time required to reproduce results with IPC=50. However, the marginal performance gains from adjusting the batch size will not diminish the 7.6% performance improvement achieved by our INFER in the IPC=50 scenario.
> > >
> > > We will include the ablation study on batch size in our revised paper as per your suggestion. We hope the experimental results address your concerns regarding our performance enhancement. We look forward to any further questions or feedback to help improve our paper!

---

> > > > ### Comment · Reviewer_RT9a · 2024-11-28
> > > > **Fairness of the performance comparison.**
> > > >
> > > > Thank you for the response. One of my primary concerns pertains to the fairness of the performance comparisons reported in the paper. As I mentioned in my previous reply, the primary baseline, SRe2L, utilizes a batch size of 1024, while the authors of this work adopt a batch size of 32. Furthermore, the configuration of G-VBSM appears to differ from the experimental setup employed in this paper. This discrepancy introduces an unfair comparison, leading me to question the results presented in Tables 1, 2, and 4 of the paper.
> > > >
> > > > Even though the authors have provided comparisons for several methods under a batch size of 128, demonstrating their method's superior performance, the evaluation still lacks a fully fair comparison against the current state-of-the-art methods under the same experimental settings. For instance, CDA does not employ Mixup or similar augmentations, whereas such techniques are used in this paper. However, the authors seem to have directly adopted the reported results of CDA from other papers, despite the evaluation settings not being entirely consistent.
> > > >
> > > > It is imperative to establish a complete experimental setup as a fundamental principle in conducting rigorous research, and the results of this paper are not convincing. Therefore, I decided to reduce my score and reject this paper.

---

> > > > > ### Author Response · Authors · 2024-11-28
> > > > > **The fairness of reviewing is also important**
> > > > >
> > > > > A **clear mistake of your claim** is:
> > > > > > For instance, CDA does not employ Mixup or similar augmentations,
> > > > >
> > > > > However, CDA [2] releases their codes and clearly states in [their repo](https://github.com/VILA-Lab/SRe2L/tree/main/CDA#relabel) saying
> > > > > > We follow SRe2L relabeling method and use the above squeezing model to relabel distilled datasets.
> > > > >
> > > > > where Sre2L have used both MixUp and CutMix in [their codes](https://github.com/VILA-Lab/SRe2L/blob/8d00cba364bfa34aa1f5a8107dc041ae1be8ebe6/SRe2L/relabel/generate_soft_label.py#L66). **This directly contradicts your claim**, which is **the only supporting argument for your concerns** regarding experimental fairness.
> > > > >
> > > > > Additionally, with the extended discussion period for ICLR2025, we kindly ask you to logically substantiate your claim:
> > > > > >the configuration of G-VBSM appears to differ from the experimental setup employed in this paper.
> > > > >
> > > > > Could you provide a clear explanation of the specific differences in the experimental setup?
> > > > >
> > > > > Your further claim:
> > > > > >  the authors seem to have directly adopted the reported results of CDA from other papers
> > > > >
> > > > > This is incorrect. As explicitly stated in our paper, we reproduced the CDA results following the state-of-the-art LPLD methodology, as you yourself mentioned. We will also release our codes after the reviewing period of ICLR. Constructive academic discussions rely on fairness, accuracy, and an open-minded approach to all presented evidence.
> > > > >
> > > > > We respectfully request that you revise your misleading arguments, as they misrepresent both our paper and our responses. With five days remaining in the discussion period, we believe there is still sufficient time to engage in constructive discussion and address your concerns, rather than prematurely concluding and refusing further discussion.
> > > > >
> > > > > Thank you.

---

> ### Comment · Reviewer_RT9a · 2024-12-01
> **Fairness of the performance comparisons reported in Tables 1, 2, and 4**
>
> Thank you for your detailed response. I acknowledge a misunderstanding of certain technical aspects regarding CDA.
>
> However, a primary concern of mine is **the fairness of the performance comparisons reported in Tables 1, 2, and 4**, as the **batch sizes employed for the comparison methods lack consistency**. For instance, in Table 1, the primary baseline, SRe2L [1], utilizes a batch size of 128 on CIFAR-100 (as stated in Table 10 on page 17 in [1]), whereas this study employs a batch size of 64 (Table 5 on page 15 in this paper). Similarly, in Table 2, **on ImageNet-1K, SRe2L [1] uses a batch size of 1024 (referenced in Table 7 on page 15 in [1]), while this study adopts a batch size of 32 (Table 5 on page 15 in this paper)**.
>
> Previous research [2, 3] and results presented by the authors indicate that **batch size significantly affects performance**. For example, using a **smaller batch size** in SRe2L, RDED, and G-VBSM can **yield notable performance improvements**. Therefore, the performance comparisons presented in this paper may be unfair.
>
> I welcome further discussion on this matter and ensure a fair evaluation framework.
>
> [1] Squeeze, Recover and Relabel: Dataset Condensation at ImageNet Scale From A New Perspective. NeurIPS, 2023
>
> [2] Dataset distillation in large data era.
>
> [3] Elucidating the Design Space of Dataset Condensation. NeurIPS, 2024.

---

> > ### Author Response · Authors · 2024-12-03
> > **Addressing Errors in Reviewer RT9a Comments**
> >
> > Thank you for your feedback. However, I must respectfully highlight **a mistake in your comments**.
> >
> > Your claim:
> > >using a smaller batch size in SRe2L, RDED, and G-VBSM can yield notable performance improvements.
> >
> > This assertion is overly absolute and does not hold universally, as evidenced by our experiments and findings from existing literature. Specifically:
> >
> > **(1) Empirical Evidence from Our Experiments**, we conducted experiments with SRe2L and G-VBSM using a batch size of 32 on the ImageNet-1k dataset:
> >
> > ImageNet-1K, **Batch size 32**, the greater number of compressed labels indicating fewer soft labels used in training, which is better.
> >
> > | ipc=10| Sre2L ($1 \times$ compressed labels）  |G-VBSM ($1 \times$ compressed labels)|INFER （$300 \times$ compressed labels） |
> > |-------- |  -------- |-------- |-------- |
> > |Smaller Batch size=32 |18.4  |30.8| **37.0**|
> > |Original Batch size|21.3 (Batch size=1024)|31.4 (Batch size=40)|NA|
> >
> > | ipc=50| Sre2L ($1 \times$ compressed labels）  |G-VBSM ($1 \times$ compressed labels)|INFER （$300 \times$ compressed labels） |
> > |-------- |  -------- |-------- |-------- |
> > |Smaller Batch size=32 |43.9  |48.7| **55.6**|
> > |Original Batch size|46.8 (Batch Size=1024)|51.8 (Batch Size=40)|NA|
> >
> > **Contrary to your claim**, the results demonstrate that smaller batch sizes do not consistently enhance performance. Instead, we observed clear performance degradation in both models when smaller batch sizes were used.
> >
> > **(2) Established Findings in Literature:**
> > Researchers with extensive experience training models on the ImageNet-1k dataset understand that there exists an optimal batch size for training. While smaller batch sizes may benefit certain setups, **it is incorrect to generalize this as a universal rule**.
> >
> > This understanding is supported by ablation studies from prior works:
> > - In Figure 6 of G-VBSM [4], a batch size of 60 yields worse performance than a batch size of 80 (31.2 vs 31.3).
> > - Similarly, Table 6 of CDA [2] shows that a batch size of 8 performs worse than a batch size of 16 (22.41 vs 22.75).
> > - Lastly, Table 20 of EDC [3] shows that a batch size of 10 performs worse than a batch size of 25 (76.0 vs 78.1).
> >
> > All the supporting evidence referenced comes from the papers you mentioned in your comments. Therefore, **it is inaccurate to assert that smaller batch sizes inherently lead to “notable performance improvements.”** Instead, the batch size should be tuned based on the model, dataset, and training framework to achieve optimal results. That exaplains why EDC [3], who claimed smaller batch size will benefit performance, but still adopt batch size =100 on their own methods.
> >
> > Additionally, the authors of SRe2L [1], CDA [2], G-VBSM [3], and EDC [4], who are experienced researchers, have already performed thorough hyperparameter tuning to determine the optimal batch size for their respective methods. Notably, these researchers adopted batch sizes of 1024 [1], 128 [2], 40 [3], and 100 [4], respectively, in their experiments. These choices directly contradict **your incorrect claim** that smaller batch sizes inherently lead to “notable performance improvements.”
> >
> > [1] Squeeze, Recover and Relabel: Dataset Condensation at ImageNet Scale From A New Perspective. NeurIPS, 2023
> >
> > [2] Dataset distillation in large data era.
> >
> > [3] Elucidating the Design Space of Dataset Condensation. NeurIPS, 2024.
> >
> > [4] Generalized large-scale data condensation via various backbone and statistical matching, CVPR, 2024.
> >
> > ---
> >
> > ## Summary of five rounds responses to Reviewer RT9Aa
> >
> > Our method, INFER, demonstrates outstanding performance when evaluated using the same batch sizes of 32 and 128, as reported in our latest two responses. These results thoroughly confirm the validity and significance of our contributions. Moreover, it is important to highlight that INFER uses only 1/300 (or 3.33%) of the soft labels compared to these methods. According to your criteria, this would represent an unfair setting for our method, as it operates with significantly fewer supervised soft labels.
> >
> > We appreciate your efforts as a reviewer for ICLR, a top-tier conference renowned for its rigorous and open review process. However, we kindly request a fair and objective evaluation of our work. Over the course of five rounds, we have diligently addressed your questions, including comparisons with the latest methods and clarifications on numerical issues.
> >
> > Your comments, rather than engaging with the core insights, theoretical contributions, and future directions of our paper, **only focus predominantly on finding new reference papers to critique our results using claims that are not substantiated**. This approach does not align with the constructive and forward-looking spirit of ICLR’s review process. We hope the Area Chair (AC) and Program Committee (PC) can take note of the thoroughness of our responses and the robustness of our contributions.

---

> ### Comment · Reviewer_RT9a · 2024-12-03
>
> I'm grateful for the detailed response and experiments. It is widely recognized in the deep learning community that smaller batch sizes do not consistently yield better performance. If this were true, batch sizes of 1 would be universally adopted.
>
> Furthermore, I have concerns regarding the following points:
>
> 1. Papers such as SRe2L and G-VBSM do not claim to have conducted comprehensive experiments to determine optimal batch sizes, contrary to the authors' assertions.
>
> 2. The impact of small batch sizes is extensively analyzed in EDC both empirically and theoretically. The paper illustrates that smaller batch sizes effectively increase the number of iterations, aiding in the prevention of model under-convergence. The authors' paper lacks a detailed empirical or theoretical analysis to substantiate this claim.
>
> 3. The authors' own experiments indicate that batch size adjustments significantly affect performance. For instance, increasing the batch size from 32 to 128 results in a performance drop from 37% to 34.85% for INFER.
>
> In conclusion, I believe the authors have selectively chosen SRe2L as the primary baseline, overlooking the potential for performance improvements in the baselines through batch size adjustments, resulting in an unfair comparison.

---

> ### Author Response · Authors · 2024-12-03
> **A rigorous attitude is a fundamental requirement for peer review**
>
> Thank you for your responses. However, we would like to gently remind you of the conclusion from the batch size experiments—requested by you but subsequently overlooked—that our method, INFER, demonstrates outstanding performance when evaluated using the same batch sizes of 32 and 128. Notably, this performance is achieved using only 1/300 (or 3.33%) of the soft labels compared to the other methods.
>
> We acknowledged the performance degradation in our third response when using a batch size of 128, as per your request. However, the conclusion of these experiments, which you also appear to have overlooked, demonstrates that **we achieved a 14.85% performance improvement** while using significantly fewer soft labels (300× compressed labels for INFER vs. 40× for LPLD). We believe **a rigorous evaluation should prioritize acknowledging the experimental results over dismissing them based on intuition**.
>
> Additionally, your claim:
> > G-VBSM do not claim to have conducted comprehensive experiments
>
> This is incorrect, as the authors conducted batch size ablation study in Figure 6.
>
> Your claim:
> > the authors have selectively chosen SRe2L as the primary baseline
>
> This is biased, as SRe2L is the first work to use a model inversion approach for synthesizing distilled datasets on ImageNet-1k. Consequently, SRe2L should be considered a foundational reference for subsequent dataset distillation research.
>
> Throughout the six rounds of discussion, you have referenced SRe2L, CDA, RDED, LPLD, and G-VBSM solely to critique the experimental results. However, you have not offered any opinions or feedback on the core insights, theoretical contributions, or foundational aspects of these related works or our submission.
>
> After we demonstrated the superiority of our proposed INFER, you repeatedly overlooked the positive conclusions supported by our experimental results. Instead, you consistently referenced additional papers to request further numerical comparisons, repeatedly shifting focus away from the demonstrated strengths of our method. However, your continuous misinterpretations of the referenced papers, including CDA and G-VBSM, suggest that you have not thoroughly read these works. Instead, **you appear to cite their experimental results solely because their values differ/are higher from ours, without considering the broader context or methodology underlying those results.** However, we demonstrated that our proposed INFER outperforms  the papers you referenced.

---

### Author Response · Authors · 2024-12-03
**Summary of our responses during the discussion period**

Dear AC and reviewers,

We would like to sincerely thank you for dedicating your valuable time and effort to providing insightful feedback and suggestions that have greatly contributed to enhancing the quality of our work.

We have carefully addressed all comments and suggestions raised by the Reviewers. The additional experiments, analyses, and discussions will be incorporated into the revised manuscript and appendix.

Below, we provide a summary of the key points discussed during the rebuttal:


- Comparison with SOTAs.
> We conducted a comprehensive comparison with SOTA methods, including RDED, CDA, LPLD, DATM, IID, and G-VBSM, and highlighted our approach’s notable advantages in both efficiency and effectiveness under fair evaluation settings. (**RT9a, hymx, enyK, aHjp**)

- Additional experiments.
> We have conducted extensive experiments to validate the cross-architecture generalization capability of our approach (**enyK**).
>  We also performed ablation experiments to analyze the detailed implementation of the initialization and optimization processes for the cross-feature compensator (**hymx**).
>  We further validated the strong efficiency of our method by applying our static labeling strategy to other SOTA methods, including SRe2L, CDA, and RDED (**RT9a, hymx, aHjp**).

- Computational cost.
> We provided a detailed comparison of computational costs with other SOTA methods. (**RT9a, aHjp**)

- Budget allocation.
> We analyzed the utilization of the distillation budget and detailed the adjustment of $P$ to ensure fairness in comparisons. (**hymx, enyK, aHjp**)

- Compression ratio calculation.
> We provided a detailed explanation of the compression ratio calculation and demonstrate the advantages of our method in compressing soft labels. (**aHjp**)

- Batch size ablation study.
> We conducted experiments to verify that our method, INFER, outperforms the baseline methods SRe2L, CDA, G-VBSM, and LPLD using the same batch sizes of 32 and 128, as requested by **RT9a**.

- Further discussion.
> We also discussed potential applications in privacy preservation, the feasibility of combining with factorization methods, current limitations, and future trends in dataset distillation. (**hymx**)


As the end of the discussion period approaches, we kindly ask if our responses have satisfactorily addressed your concerns. Your feedback would be greatly appreciated, and we would be delighted to engage in further discussions if needed.

Sincerely,

The Authors

---

### Meta-Review · Area_Chair_JTGW · 2024-12-22

**Metareview:**

This work proposes a new framework for dataset distillation that can better utilise the distillation budget and consider inter-class feature distributions. The idea is to move from the traditional class-specific data-label framework and develop a one-instance-for-all-class approach. The key components consist of the universal feature compensator and the enhancement of synthetic data with inter-class augmentation. Experimental study is conducted on benchmark datasets to verify the efficacy of the proposed method.

Reviewers comment that the proposed method can significantly reduce the storage, the paper is well written, the work is technically sound with thorough evaluations, and the proposed inter-class feature compensator is novel, and so on. Meanwhile, the reviewers raise concerns related to the comparison with more methods, the evaluation of computational cost, the fairness of experimental study, the privacy issue related to the use of real instances, the effectiveness under high compression ratios, the cross-architecture evaluation, the generality of the proposed idea, and the clarification of details. The authors provide a rebuttal of high quality. The responses to the comments of three reviewers address most or all of the concerns and lead to the increase of score. The authors’ responses during the multiple rounds of discussion with Reviewer RT9a are effective. During the further discussion between reviewers, AC highlights two concerns for them to check, including 1) the fairness of the experimental comparison (raised primarily by Reviewer RT9a), and 2) the effectiveness and consistency of the proposed idea under high compression ratios (raised primarily by Reviewer hymx). Reviewer aHjp replies that the two concerns have been well addressed during the rebuttal period. The final ratings are 3 (Reviewer RT9a), 5 (Reviewer hymx), 6 (Reviewer enyK) and 8 (Reviewer aHjp) .

AC has carefully reviewed the submission, along with the reviews, the rebuttals, the discussions, and the message provided by the authors. AC concurs with the reviewers regarding the novelty and efficiency of the proposed method and are satisfied with the authors' rebuttal in addressing the major concerns. Taking all the factors into account, AC recommends to accept this work and suggests the authors to further strengthen this work by properly incorporating the responses in the rebuttal.

**Additional Comments On Reviewer Discussion:**

The reviewers raise concerns related to the comparison with more methods, the evaluation of computational cost, the fairness of experimental study, the privacy issue related to the use of real instances, the effectiveness under high compression ratios, the cross-architecture evaluation, the generality of the proposed idea, and the clarification of details. The authors provide a rebuttal of high quality. The responses to the comments of three reviewers address most or all of the concerns and lead to the increase of score. The authors’ responses during the multiple rounds of discussion with Reviewer RT9a are effective. During the further discussion between reviewers, AC highlights two concerns for them to check, including 1) the fairness of the experimental comparison (raised primarily by Reviewer RT9a), and 2) the effectiveness and consistency of the proposed idea under high compression ratios (raised primarily by Reviewer hymx). Reviewer aHjp replies that the two concerns have been well addressed during the rebuttal period. The final ratings are 3 (Reviewer RT9a), 5 (Reviewer hymx), 6 (Reviewer enyK) and 8 (Reviewer aHjp) .

AC has carefully reviewed the submission, along with the reviews, the rebuttals, the discussions, and the message provided by the authors. AC concurs with the reviewers regarding the novelty and efficiency of the proposed method and are satisfied with the authors' rebuttal in addressing the major concerns. Taking all the factors into account, AC recommends to accept this work.

---

### Decision · Program_Chairs · 2025-01-22

Accept (Poster)